# LINKING PROCESS TO OUTCOME: CONDITIONAL RE-WARD MODELING FOR LLM REASONING

**Zheng Zhang[1,2], Ziwei Shan[1], Kaitao Song[3], Yexin Li[2]\*, Kan Ren[1]\***

[1]School of Information Science and Technology, ShanghaiTech University
[2]State Key Laboratory of General Artificial Intelligence, BIGAI
[3]Independent Researcher
{zhangzheng2024,shanzw2022,renkan}@shanghaitech.edu.cn
liyexin@bigai.ai
stillkeeptry@outlook.com

## ABSTRACT

Process Reward Models (PRMs) have emerged as a promising approach to en-hance the reasoning capabilities of large language models (LLMs) by guiding their step-by-step reasoning toward a final answer. However, existing PRMs either treat each reasoning step in isolation, failing to capture inter-step depen-dencies, or struggle to align process rewards with the final outcome. Conse-quently, the reward signal fails to respect temporal causality in sequential rea-soning and faces ambiguous credit assignment. These limitations make down-stream models vulnerable to reward hacking and lead to suboptimal perfor-mance. In this work, we propose Conditional Reward Modeling (CRM) that frames LLM reasoning as a temporal process leading to a correct answer. The reward of each reasoning step is not only conditioned on the preceding steps but also explicitly linked to the final outcome of the reasoning trajectory. By enforcing conditional probability rules, our design captures the causal relation-ships among reasoning steps, with the link to the outcome allowing precise at-tribution of each intermediate step, thereby resolving credit assignment ambi-guity. Further, through this consistent probabilistic modeling, the rewards pro-duced by CRM enable more reliable cross-sample comparison. Experiments across Best-of-N sampling, beam search and reinforcement learning demonstrate that CRM consistently outperforms existing reward models, offering a principled framework for enhancing LLM reasoning. In particular, CRM is more robust to reward hacking and delivers stable downstream improvements without rely-ing on verifiable rewards derived from ground truth. The project website is at https://foundation-model-research.github.io/CRM.

## 1 INTRODUCTION

Recent advances in enhancing reasoning abilities have greatly improved the performance of large language models (LLMs) (Snell et al., 2025; Jaech et al., 2024), where models derive final answers through explicit step-by-step reasoning. Beyond prompt-based approaches (Wei et al., 2022; Diao et al., 2024; Fan et al., 2025), state-of-the-art systems like DeepSeek-R1 (Guo et al., 2025a) have further advanced reasoning capacity through reinforcement learning (RL) with verifiable rewards.

However, verifiable rewards rely on checking model outputs against ground-truth labels, whose acquisition is costly and difficult to scale for general reasoning improvements. Reward models provide a promising alternative by extrapolating reward signals across general LLM reasoning pro-cesses. Broadly, these models fall into two categories: outcome reward models (ORMs) (Cobbe et al., 2021; Yu et al., 2024), which provide feedback at the final step; and process reward models (PRMs) (Wang et al., 2024; Li & Li, 2025; Yuan et al., 2025), which provide finer-grained rewards at the level of individual reasoning steps or even tokens.

---

\*Corresponding Author.

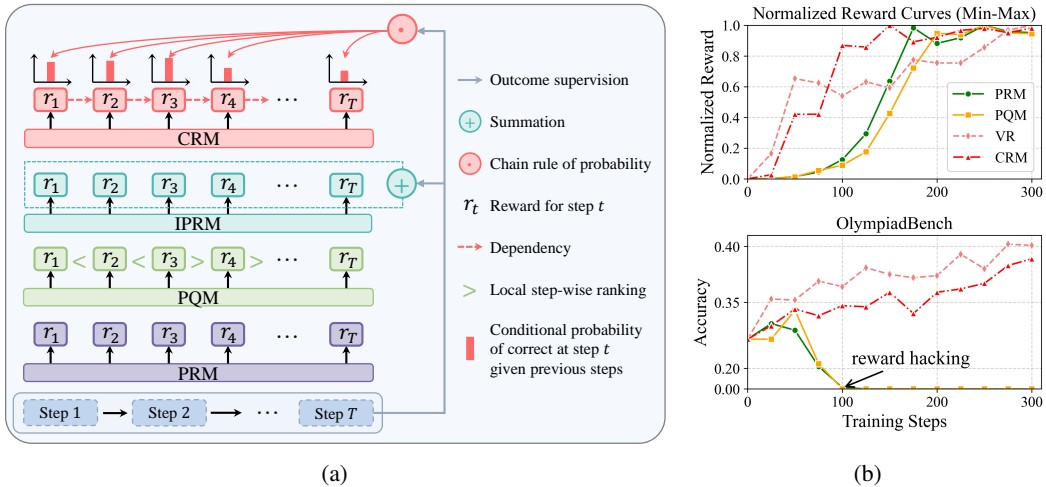

(a)          (b)

Figure 1: (a) Comparison of reward modeling paradigms. Our CRM explicitly conditions each step reward on the previous reasoning steps and aligns it with the final outcome. (b) RL training with different reward models. Our CRM is more robust to reward hacking and achieves performance on par with training using verifier rewards (VR).

Despite offering nuanced signals for reasoning processes, existing PRMs face several limitations. (i) *Isolated step modeling*: As shown in Figure 1a, most PRMs (Lightman et al., 2023; Wang et al., 2024; Luo et al., 2024; Shao et al., 2024) assess each reasoning step in isolation, neglecting the intrinsic sequential dependencies of reasoning. (ii) *Limited outcome awareness*: While some methods (Yu et al., 2024; Li & Li, 2025; Yuan et al., 2025) attempt to mitigate isolated step rewards, they often fail to effectively link step-wise rewards with the outcome. For example, PQM (Li & Li, 2025), illustrated in Figure 1a, relies on relative comparisons between neighboring steps (e.g., greater or smaller reward), but lacks explicit modeling of the final outcome, which may result in biased process rewards. IPRM (Yuan et al., 2025) parameterizes the outcome reward as the logarithmic sum of process rewards, yet it fails to capture how a specific step relates to the final outcome and lacks nuanced modeling of inter-step dependencies. This leads to ambiguous credit assignment from the final outcome back to the intermediate reasoning steps. As a result, existing methods are prone to reward hacking, where rewards continue to increase while the actual task accuracy declines, which we have observed in our experiments and has been shown in Figure 1b.

To address these limitations, we propose Conditional Reward Modeling (CRM), which frames reasoning as *a temporal process through which an LLM progressively approaches the correct final answer*. In our formulation, the reward signal is modeled as the probabilistic evolution of deriving the correct outcome conditioned on the existing reasoning steps. At each step, the process reward is treated as a conditional probability dependent on all the preceding steps, thereby capturing causal structure inherent in sequential reasoning. Furthermore, by explicitly linking each process reward to the final outcome via the conditional probability chain rule, CRM enables precise attribution of the final result to individual reasoning steps, effectively resolving the credit assignment ambiguity prevalent in prior work. An additional advantage is that the probabilistically consistent formulation of process reward signals across reasoning trajectories facilitates cross-sample comparison, which significantly benefits downstream tasks such as Best-of-N sampling, beam search, and RL optimization. As shown in Figure 1a, CRM jointly models inter-step dependencies and the relationship between intermediate rewards and the final outcome.

Our contributions are three-fold as follows. (i) *Conditional reward modeling framework*: We introduce CRM, which defines each step's reward as a conditional probability dependent on all preceding steps, thereby capturing inter-step dependencies. (ii) *Precise credit assignment*: By linking process rewards to the final outcome, CRM resolves the ambiguity of credit assignment in existing PRMs. (iii) *Practical effectiveness and robustness*: CRM enhances cross-sample comparison and improves various downstream tasks. Experiments demonstrate that our CRM achieves superior performance, remains robust to reward hacking and delivers stable reasoning improvements without reliance on a verifier that uses ground-truth labels.

## 2 RELATED WORK

**Enhancing the Reasoning Ability of LLMs.** To enhance LLM reasoning, prior studies have explored test-time search (e.g., Best-of-N, beam search) with reward models to select promising solutions (Xie et al., 2023; Zhang et al., 2025a; Snell et al., 2025), and post-training via RL (Jaech et al., 2024; Guo et al., 2025a). Recent work (Zeng et al., 2025b; Guo et al., 2025a) has investigated RL with verifiable rewards for reasoning tasks. However, these approaches heavily rely on ground-truth and thus do not easily scale. Other studies (Cui et al., 2025; Cheng et al., 2025) integrate reward models into RL to provide dense feedback, but reward hacking remains a major concern (Gao et al., 2024).

**Process Reward Model.** Most existing studies (Lightman et al., 2023; Wang et al., 2024; Luo et al., 2024; Shao et al., 2024) treat PRM as step-level classification, which treats steps in isolation and ignores inter-step dependencies. Recent studies (Lu et al., 2024; Li & Li, 2025; Yuan et al., 2025) attempt to move beyond classification-based PRMs. Li & Li (2025) reframes PRM as a Q-value ranking problem to model the relationships between steps, but overlooks explicit modeling of the final outcome, resulting in a gap between process rewards and the final outcome. Yuan et al. (2025) instead proposes a parameterized formulation of outcome, leveraging outcome labels to train the PRM, but the relationship between intermediate step rewards and the final outcome remains unclear, and the lack of step dependency modeling leads to ambiguous credit assignment. Another line of work (Chen et al., 2025; Guo et al., 2025b; She et al., 2025; Zhao et al., 2025) integrates reasoning capabilities into reward modeling, such as R-PRM (She et al., 2025) and GenPRM (Zhao et al., 2025), which evaluate reasoning processes with detailed thinking. However, these approaches often involve complex training procedures and incur high inference costs.

## 3 METHODOLOGY

### 3.1 TASK DESCRIPTION

Given a question $x$, LLM $\pi$ generates a response $y = (a_1, a_2, \ldots, a_T)$ where $a_t$ denotes the $t$-th reasoning step and $T$ is the total step number. Let $a_{\leq t}$ denote the sequence of the first $t$ steps. Each response is assigned a binary label $l \in \{0, 1\}$ indicating whether the final answer derived from it is correct.

We model multi-step reasoning as a finite-horizon Markov Decision Process (MDP) $\mathcal{M} = (\mathcal{S}, \mathcal{A}, \mathcal{P}, r, \gamma)$ driven by an autoregressive LLM $\pi$ acting as the policy. $\mathcal{S}$ is the state space, $\mathcal{A}$ is the action space, $\mathcal{P}$ represents transition dynamics, $r : \mathcal{S} \times \mathcal{A} \to \mathbb{R}$ is the reward function, and $\gamma \in [0, 1]$ is the discount factor. At reasoning step $t$, the state is defined as $s_t = (x, a_{\leq t-1})$, which contains the question $x$ and the sequence of previously generated reasoning steps $a_{\leq t-1}$. The action $a_t$ is a reasoning step generated based on the state $s_t$. The state transition is deterministic, as the next state is uniquely determined by concatenating the previous sequence with the current output $a_t$. $r_t$ denotes the reward provided by the environment or generated by the reward model for action $a_t$.

### 3.2 CRM MODELING

To remedy isolated step modeling and limited outcome awareness in existing PRMs, we model LLM reasoning as a temporal process in which the probability of reaching the correct answer evolves with step $t$. However, in practice, it is difficult to directly quantify the extent to which reasoning is approaching the correct answer. Instead, we choose to model the complementary event: the reasoning process entering a wrong state, which implies that the reasoning trajectory can no longer yield the correct answer. Formally, we define $z$ as the index of the first step at which the reasoning process enters such a wrong state, with $z \geq 1$. If no wrong state occurs throughout the trajectory, then $z > T$, meaning the reasoning is correct and the final answer is correct ($l = 1$). Conversely, if the final answer is incorrect ($l = 0$), then the trajectory has entered a wrong state during reasoning, with $z \leq T$. Let $p(z; x, a_{\leq z})$ denote the probability mass function of a wrong state occurring at step $z$. Accordingly, the probability that a wrong state has already occurred at or before step $t$ is

$$W(t; x, a_{\leq t}) = \Pr(z \leq t) = \sum_{z=1}^{t} p(z; x, a_{\leq z}) \tag{1}$$

The probability of maintaining correct reasoning up to step $t$ is its complement,

$$S(t; x, a_{\leq t}) = \Pr(z > t) = 1 - W(t; x, a_{\leq t}) = \sum_{z=t+1}^{\infty} p(z; x, a_{\leq t}) \tag{2}$$

The discrete probability mass function of the wrong state occurring at step $t$ is

$$p(t; x, a_{\leq t}) = \Pr(z = t) = W(t; x, a_{\leq t}) - W(t - 1; x, a_{\leq t-1}) \tag{3}$$

For simplicity, we use $p(t)$, $W(t)$, $S(t)$ to denote $p(t; x, a_{\leq t})$, $W(t; x, a_{\leq t})$ and $S(t; x, a_{\leq t})$.

A reasoning trajectory is inherently causal: the correctness of step $t$ logically depends on all preceding $(t-1)$ steps. Building on this view, we adopt a conditional probability perspective and derive $h(t)$, the probability that step $t$ enters a wrong state given that all previous $(t-1)$ steps were correct.

$$h(t) = \Pr(z = t | z \geq t) = \frac{\Pr(z = t)}{\Pr(z \geq t)} = \frac{p(t)}{S(t-1)} \tag{4}$$

Naturally, $1 - h(t)$ corresponds to the complementary event: the probability that the current step is correct given that all previous $(t-1)$ steps were correct. This formulation explicitly captures step dependencies through conditional probabilities, addressing the limitations of prior studies (Lightman et al., 2023; Wang et al., 2024; Luo et al., 2024; Shao et al., 2024) that overlook inter-step relations.

To explicitly link intermediate steps to the final outcome, we first apply the chain rule of probability to establish the relationships among $S(t)$, $W(t)$, $p(t)$, and $h(t)$.

$$S(t) = \Pr(z > t) = \Pr(z \neq 1, z \neq 2, ..., z \neq t) = \Pr(z \neq 1) \cdot \Pr(z \neq 2 | z \neq 1) \cdots$$

$$\cdots \Pr(z \neq t | z \neq 1, z \neq 2..., z \neq t - 1) = \prod_{k=1}^{t} [1 - \Pr(z = k \mid z \geq k)] = \prod_{k=1}^{t} (1 - h(k)) \tag{5}$$

$$W(t) = \Pr(z \leq t) = 1 - S(t) = 1 - \prod_{k=1}^{t} (1 - h(k)) \tag{6}$$

$$p(t) = \Pr(z = t) = h(t) \prod_{k=1}^{t-1} (1 - h(k)) \tag{7}$$

For the final step of the reasoning trajectory $T$, we have $S(T) = \prod_{t=1}^{T} (1 - h(t))$ from Eq. 5, where $S(T)$ reflects the probability that the reasoning process reaches the correct final answer. We next seek a dense, step-wise reward signal aligned with this outcome probability.

We apply Potential-Based Reward Shaping (PBRS) (Ng et al., 1999) to the task of multi-step reasoning. PBRS proposes that a dense reward function $R'$ can be constructed from an original sparse reward function $R$ by adding a shaping term derived from a potential function $\Phi(s)$ defined over the state space. For a transition from state $s_t$ to $s_{t+1}$, the shaped reward is given by:

$$R'(s_t, a_t, s_{t+1}) = R(s_t, a_t) + \gamma\Phi(s_{t+1}) - \Phi(s_t) \tag{8}$$

PBRS guarantees that any policy optimal under the shaped reward $R'(s_t, a_t, s_{t+1})$ is also optimal under the original reward $R(s_t, a_t)$. Thus, using the shaped reward guides the learning process more effectively without changing the fundamental goal of the task. The key to applying PBRS lies in selecting an appropriate potential function $\Phi(s_t)$ which is effective when it encodes goal-directed progress (Lo et al., 2024; Müller & Kudenko, 2025; Mindom et al., 2021; Wiewiora et al., 2003). We define the potential function as $\log S(t)$, which denotes the log-likelihood of eventually reaching a correct answer from the current state, thereby explicitly capturing the progress toward the final correct answer. Consequently, substituting this definition of $\log S(t)$ into the potential function yields the following explicit form:

$$\Phi(s_t) \equiv \log S(t) = \log \left( \prod_{k=1}^{t} (1 - h(k)) \right) = \sum_{k=1}^{t} \log(1 - h(k)) \tag{9}$$

We now apply the general PBRS formula (Eq. 8) to derive the specific process reward $r_t$, which corresponds to the transition from state $s_{t-1}$ to $s_t$.

$$r_t \equiv R'(s_{t-1}, a_{t-1}, s_t) = R(s_{t-1}, a_{t-1}) + \gamma\Phi(s_t) - \Phi(s_{t-1}) = \log(1 - h(t)) \tag{10}$$

where original reward $R = 0$ for all intermediate steps, and the discount factor is fixed at $\gamma = 1$.

The above derivation yields a dense and accurate process reward, $r_t = \log(1 - h(t))$, which serves as a credit assignment scheme grounded in our modeling. The probability that the reasoning process reaches the correct final answer, $S(T)$, satisfies $S(T) = \prod_{t=1}^{T}(1 - h(t)) = \prod_{t=1}^{T} e^{r_t}$. Through this decomposition, we explicitly link process rewards to the outcome, addressing the limitation of prior work (Yu et al., 2024; Li & Li, 2025; Yuan et al., 2025) that lacks explicit modeling of the relationship between process and outcome.

### 3.3 TRAINING OF CONDITIONAL REWARD MODEL

Let $\mathcal{D} = \{(x_i, y_i, l_i, z_i)\}$ denote the training dataset, where $x_i$ is the $i$-th question, $y_i$ is the corresponding response (multi-step reasoning trajectory), $l_i$ indicates whether the final answer is correct, and $z_i$ is the index of the first wrong state. We use a large language model $f_\phi$ with parameters $\phi$ to initialize the reward model. Since both $S(T)$ and the process reward $r_t$ are functions of $h(t)$, we train the model to predict $h(t)$. At the $t$-th step of the reasoning trajectory, the model takes the question $x$ and the first $t$ reasoning steps as input, and predicts the $h(t) = f_\phi(x, a_{\leq t})$.

For samples where the reasoning process reaches the correct final answer ($l_i = 1$), we maximize $S(T)$ and the corresponding loss $\mathcal{L}_S$ is:

$$\mathcal{L}_S(x_i, y_i) = -\log \Pr(z_i > T) = -\log S(T) = -\log\left[\prod_{t=1}^{T}(1 - h(t))\right] \quad (11)$$

For samples where the reasoning trajectory fails to reach the correct final answer ($l_i = 0$), we minimize $S(T)$. Moreover, since the reasoning process enters a wrong state at step $z_i$, we encourage the model to identify this step by maximizing $p(z_i)$, the probability of a wrong state occurring exactly at step $z_i$. The loss $\mathcal{L}_W$ and $\mathcal{L}_z$ for this case are as follows:

$$\mathcal{L}_W(x_i, y_i) = -\log \Pr(z_i \leq T) = -\log(1 - S(T)) = -\log\left[1 - \prod_{t=1}^{T}(1 - h(t))\right] \quad (12)$$

$$\mathcal{L}_z(x_i, y_i, z_i) = -\log \Pr(z_i) = -\log p(z_i) = -\log\left[h(z_i)\prod_{t=1}^{z_i-1}(1 - h(t))\right] \quad (13)$$

Figure 2 illustrates the roles of the three loss terms. The overall loss to be minimized is as follows:

$$\mathcal{L} = \frac{1}{|\mathcal{D}|}\sum_{i=1}^{|\mathcal{D}|}\left[l_i\,\mathcal{L}_S(x_i, y_i) + (1 - l_i)\big(\mathcal{L}_W(x_i, y_i) + \mathcal{L}_z(x_i, y_i, z_i)\big)\right] \quad (14)$$

This consistent probabilistic modeling and training ensure that CRM is grounded in clear probabilistic semantics: for example, the value of $S(t)$ at any step $t$ across different samples consistently carries the same probabilistic meaning, allowing them to be compared under the same scale. By contrast, previous approaches (Li & Li, 2025; Yuan et al., 2025) struggle to achieve accurate cross-sample comparison (see Appendix E for detailed analysis).

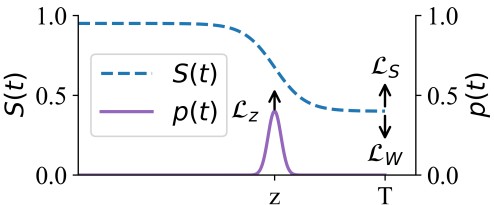

Figure 2: Effects of three loss terms.

## 4 EXPERIMENTS

### 4.1 EXPERIMENTAL SETTINGS

**Reward Model Training.** We train the reward model on the Math-Shepherd dataset (Wang et al., 2024), which integrates questions from GSM8K (Cobbe et al., 2021) and MATH (Hendrycks et al., 2021) and provides responses annotated with step-level process labels. Following prior research (Li & Li, 2025; Guan et al., 2025), we augment the pre-trained model with a value head. The reward models are trained using full parameter fine-tuning. Further details can be found in Appendix A.

**Baselines and Evaluation.** We compare our CRM against representative baselines, including ORM, vanilla PRM (Wang et al., 2024), PQM (Li & Li, 2025) and IPRM (Yuan et al., 2025). For fairness, all baselines are re-implemented within the same pipeline, backbone, and training data. We then leverage the rewards provided by the reward model to (i) select an optimal response in Best-of-N (Section 4.2), (ii) guide beam search (Section 4.3), and (iii) optimize LLM reasoning through RL (Section 4.4).

## 4.2 BEST-OF-N SAMPLING EXPERIMENTS

Best-of-N sampling generates $N$ responses for a given question and selects the optimal one using a reward model, evaluating the model's ability to identify correct samples at the trajectory level. Our CRM computes $S(T)$ as the trajectory-level score, and we evaluate it on two popular math reasoning datasets: GSM-Plus (Li et al., 2024) and MATH500 (Lightman et al., 2023).

Table 1: Best-of-N accuracy across models. **Bold** and underlined values denote the top two results.

| Models | Methods | GSM-Plus | | | | | MATH500 | | | | |
|---|---|---|---|---|---|---|---|---|---|---|---|
| | | @8 | @16 | @32 | @64 | @128 | @8 | @16 | @32 | @64 | @128 |
| | Major@N | 66.3 | 65.6 | 64.7 | 64.2 | 64.2 | 46.2 | 46.0 | 45.2 | 45.0 | 44.8 |
| | Pass@N (Oracle) | 79.2 | 81.2 | 82.2 | 83.3 | 85.5 | 73.8 | 79.2 | 84.6 | 87.4 | 90.2 |
| Qwen2.5-3B-Instruct | ORM | 66.8 | 67.2 | 66.4 | 65.7 | 65.7 | 51.6 | 51.4 | 51.8 | 49.0 | 49.2 |
| | PRM | 67.6 | 67.9 | 67.7 | 66.9 | 66.7 | **54.2** | 55.2 | 55.2 | 54.2 | 54.6 |
| | PQM | **68.5** | **69.2** | **68.5** | 68.2 | 68.0 | 53.2 | 54.4 | 54.8 | 54.8 | 55.8 |
| | IPRM | 65.5 | 66.2 | 66.8 | 66.5 | 66.2 | 52.4 | 52.0 | 52.0 | 52.2 | 53.0 |
| | **CRM (ours)** | 67.8 | 68.6 | 67.9 | **68.4** | **68.7** | 53.0 | **56.4** | **56.6** | **55.8** | **56.6** |
| LLaMA3.1-8B | ORM | 66.9 | 67.4 | 67.1 | 67.2 | 66.6 | 47.4 | 46.4 | 44.6 | 45.2 | 45.6 |
| | PRM | **67.9** | 68.2 | 68.5 | **68.8** | **68.9** | 48.0 | 48.0 | 49.8 | 49.0 | 47.6 |
| | PQM | 66.4 | 67.0 | 66.2 | 66.5 | 67.2 | **51.0** | **51.4** | 48.8 | 49.0 | 48.4 |
| | IPRM | 65.1 | 64.3 | 63.9 | 63.5 | 63.7 | 48.4 | 45.8 | 44.2 | 46.0 | 45.8 |
| | **CRM (ours)** | 67.8 | **68.8** | **69.1** | 68.6 | 68.5 | 49.4 | 50.6 | **50.6** | **49.8** | **50.6** |

**CRM demonstrates stronger trajectory-level selection in Best-of-N sampling.** As shown in Table 1, CRM consistently ranks at or near the top across both datasets and model families. For example, on MATH500 with Qwen2.5-3B-Instruct, CRM reaches 56.6% at $N$=32, surpassing the strongest baseline by +1.4 and remaining ahead across $N$=16, 64, and 128. By conditioning each step's reward on all preceding steps, CRM ensures that the trajectory score reflects the coherence of the entire reasoning chain rather than isolated fragments. This

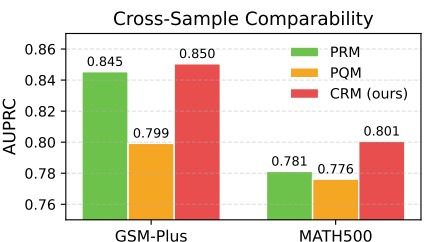

Figure 3: Cross-sample comparability.

holistic formulation enables CRM to assess logical consistency across steps and more reliably distinguish correct trajectories from superficial ones, leading to stronger trajectory-level selection. A more in-depth analysis of the performance characteristics of BoN is provided in Appendix I.

**CRM shows superior cross-sample comparability.** Best-of-N evaluation focuses on the model's ability to identify the correct response among $N$ responses for the same question, where our CRM already shows clear advantages. To move beyond this setting and assess cross-question comparability, we mix all responses from different questions and adopt AUPRC (Area Under the Precision–Recall Curve) as the metric, which measures whether correct responses are concentrated among those with higher rewards. A higher AUPRC indicates that when ranking all responses globally by reward score, correct responses are consistently placed toward the top. As shown in Figure 3, our CRM outperforms the baselines, validating its superior cross-sample comparability. This advantage arises from the consistent probabilistic modeling (see Section 3.3), which ensures that reward signals carry the same semantic meaning across reasoning trajectories, thereby making them directly comparable across samples.

Table 2: Beam Search accuracy on MATH500 and Gaokao2023.

| Models | Methods | MATH500 | | | | GAOKAO2023 | | | |
|---|---|---|---|---|---|---|---|---|---|
| | | $N=4$ | $N=8$ | $N=20$ | $N=100$ | $N=4$ | $N=8$ | $N=20$ | $N=100$ |
| Qwen2.5-Math-1.5B | ORM | 50.73 | 54.80 | 56.80 | 58.07 | 35.58 | 38.18 | 38.44 | 40.17 |
| | PRM | 51.80 | 55.73 | 56.87 | 58.00 | 34.72 | 37.84 | 38.70 | 38.96 |
| | PQM | 52.67 | 56.60 | 58.87 | 58.80 | 36.88 | 38.61 | 40.61 | 39.83 |
| | IPRM | 44.27 | 47.27 | 48.33 | 47.47 | 32.55 | 34.46 | 35.32 | 34.55 |
| | **CRM (ours)** | **54.07** | **58.40** | **61.00** | **63.00** | **38.70** | **39.74** | **41.04** | **43.55** |
| Qwen2.5-Math-7B | ORM | 51.87 | 57.67 | 59.47 | 60.73 | 37.49 | 40.26 | 44.07 | 43.72 |
| | PRM | 52.13 | 55.67 | 59.93 | 60.13 | 37.58 | 40.52 | 41.04 | 43.81 |
| | PQM | 52.73 | 57.87 | 59.20 | 61.13 | 37.84 | 40.61 | 42.60 | 43.29 |
| | IPRM | 49.53 | 54.07 | 54.20 | 52.60 | 35.67 | 38.53 | 40.26 | 39.57 |
| | **CRM (ours)** | **56.07** | **60.60** | **62.87** | **64.07** | **39.83** | **42.77** | **46.49** | **48.40** |
| Llama3.1-8B | ORM | 38.13 | 38.80 | 38.93 | 36.67 | 25.63 | 26.93 | **29.35** | 27.62 |
| | PRM | 37.87 | 39.67 | 40.13 | 39.53 | 26.84 | **28.66** | 27.97 | 27.97 |
| | PQM | 39.20 | 40.47 | 41.00 | **41.27** | 26.58 | 27.71 | 28.31 | 28.23 |
| | IPRM | 37.07 | 38.67 | 37.13 | 34.13 | 26.49 | 25.89 | 26.15 | 24.42 |
| | **CRM (ours)** | **40.20** | **41.00** | **42.07** | 41.00 | **26.93** | 28.40 | 28.74 | **29.96** |

## 4.3 BEAM SEARCH EXPERIMENTS

We conduct beam search on MATH500 and an out-of-domain (OOD) dataset Gaokao2023 (Liao et al., 2024) to evaluate the capacity of reward models in guiding LLM reasoning via step-level rewards. For each question, beam search initiates by sampling $N$ responses. Subsequently, a beam of $b$ candidates with the highest rewards is maintained and expanded during the generation process. Our CRM computes $S(t)$ as the step-level reward. We report the best accuracy achieved across various total sampling sizes $N$, with the results averaged over three random seeds. Detailed experimental settings are provided in Appendix A.

**CRM provides effective and consistent step-level guidance for beam search.** As shown in Table 2, our CRM achieves the highest accuracy on both datasets when using the Qwen2.5-Math-1.5B and Qwen2.5-Math-7B and demonstrates the best performance in the majority of cases for the Llama3.1-8B. Notably, the performance of CRM scales effectively with the total sampling size $N$. As $N$ increases from 4 to 100, the performance gap over baseline methods widens, underscoring the scalable advantage of CRM in selecting more promising intermediate steps within larger search spaces. This can be attributed to how the reward model guides the beam search algorithm. Specifically, beam search relies on the reward model to prune a vast number of trajectories by performing two distinct types of comparison: (i) ranking trajectories that share a common prefix, and (ii) ranking entirely distinct reasoning paths (cross-sample). By framing the reward as the conditional probability of reaching the correct answer given current partial trajectory, our consistent probabilistic modeling provides meaningful step-level rewards applicable to both ranking scenarios, thereby effectively guiding the reasoning process.

## 4.4 RL OPTIMIZATION EXPERIMENTS

Table 3: Pass@1 accuracy evaluated on six mathematical reasoning benchmarks.

| VR from outcome ground-truth | Method | MATH 500 | Minerva Math | Olympiad Bench | AIME25 | AIME24 | AMC23 |
|---|---|---|---|---|---|---|---|
| VR Disabled | PURE | 76.0 | 30.8 | 36.7 | 13.3 | 26.6 | **70.0** |
| | PRM | 71.6 | 36.3 | 32.5 | 13.3 | 10.0 | 57.5 |
| | PQM | 72.0 | 34.1 | 34.3 | 13.3 | 13.3 | 52.5 |
| | **CRM (ours)** | **77.8** | **40.0** | **39.3** | **23.3** | **43.3** | 67.5 |
| VR Enabled | Prime | 81.2 | 29.4 | 40.8 | 16.6 | 26.6 | **72.5** |
| | PURE | **82.4** | 40.0 | 41.3 | 23.3 | 23.3 | 70.0 |
| | VR | 76.2 | 38.6 | 38.0 | 16.6 | 30.0 | 62.5 |
| | **CRM + VR** | 80.4 | **43.0** | **42.1** | **26.6** | 33.3 | 72.5 |

During RL optimization, the reward model provides step-level dense rewards to guide the policy model toward generating improved reasoning trajectories. This setting allows us to empirically validate the effectiveness and accuracy of credit assignment delivered by the reward model in practice.

Our CRM provides step-wise process rewards $r_t = \log(1 - h(t))$. This reward formulation is not only theoretically justified in Section 3.2 but also empirically validated in Appendix G. We adopt RLOO (Leave-One-Out) (Ahmadian et al., 2024) to estimate the advantage based on $r_t$. Since RLOO is originally defined at the sequence level, following prior work (Cheng et al., 2025), we employ its token-level variant. We use Orz-Math-57k (Hu et al., 2025) as the RL training set. All policy models are initialized from Qwen2.5-Math-7B and trained with full parameter optimization. Details of advantage estimation, policy updates, and implementation settings are provided in Appendix A.

We compare against PRM and PQM, and additionally include two stronger RL baselines that use dense rewards: Prime (Cui et al., 2025) and PURE (Cheng et al., 2025). Prime adopts online reward model updates, jointly optimizing the reward model and policy, but its reliance on verifiers prevents adaptation to ground-truth–free settings. PURE adopts min-form credit assignment by defining the value function as the minimum of future rewards. However, as stated in its original paper, it becomes prone to reward hacking without verifiable rewards. In addition, we also add the VR baseline, where the policy is trained using only final-answer verifiable rewards. To comprehensively evaluate performance, we use six widely adopted benchmarks: AIME25 (MAA, 2025), AIME24 (MAA, 2024), AMC23 (MAA, 2023), MATH500 (Hendrycks et al., 2021), Minerva Math (Lewkowycz et al., 2022), and OlympiadBench (He et al., 2024). We report the pass@1 accuracy under the zero-shot setting.

**CRM boosts RL performance without VR.** CRM-based RL functions without verifiable rewards **(VR Disabled)**. For fair comparison with VR-dependent methods, we additionally report results where rewards from CRM are combined with verifiable rewards **(VR Enabled)**. The results are shown in Table 3. In the VR Disabled setting, CRM attains the best Pass@1 accuracy on the majority of benchmarks, significantly outperforming the baselines. For instance, on AIME24 it reaches 43.3%, exceeding PURE by +16.7. CRM consistently outperforms the VR baseline. When augmented with VR, CRM + VR yields further gains and achieves the highest performance on most benchmarks. This suggests that the process rewards provided by CRM are complementary to ground-truth–based verifiable rewards, rather than redundant. Remarkably, even without VR, CRM delivers performance comparable to VR-enabled methods. The strength of CRM lies in its explicit linkage between process rewards and outcomes, which ensures that the contribution of each step is causally aligned with the final result. Since process rewards quantify how much each step advances or undermines the probability of reaching the correct final answer, rather than relying on local heuristics, this design enables precise credit assignment. As a result, the dense and reliable reward signal facilitates effective RL optimization and remains robust to reward hacking, as further analyzed in Section 4.5.1.

## 4.5 EXTENDED ANALYSIS

In this section, we conduct a more in-depth analysis of CRM by investigating the following research questions. **RQ1**: What does reward hacking manifest as, and how can it be mitigated? **RQ2**: Does CRM exhibit self-reflection behavior? **RQ3**: How efficient is CRM in utilizing supervision data? **RQ4**: Can CRM be applied to domains beyond mathematics?

### 4.5.1 REWARD HACKING IN RL OPTIMIZATION

In RL optimization, employing a reward model is prone to reward hacking (Gao et al., 2024; Cheng et al., 2025). We observe that reward hacking typically manifests as the generation of excessively repetitive content. To capture this behavior, we introduce the repeat score, an n-gram–based (Li et al., 2016) metric ranging from 0 to 1, where higher values indicate a greater degree of textual repetition. The formal definition and examples of reward hacking are provided in Appendix C.

**Finding 1: Reward hacking manifests as a rapid reward increase accompanied by repetitive outputs and declining downstream performance (RQ1), whereas CRM remains robust.** Figure 4 reveals that under PRM and PQM, the reward quickly escalates despite declining downstream accuracy. This discrepancy arises from a pathological incentive, where models inflate re-

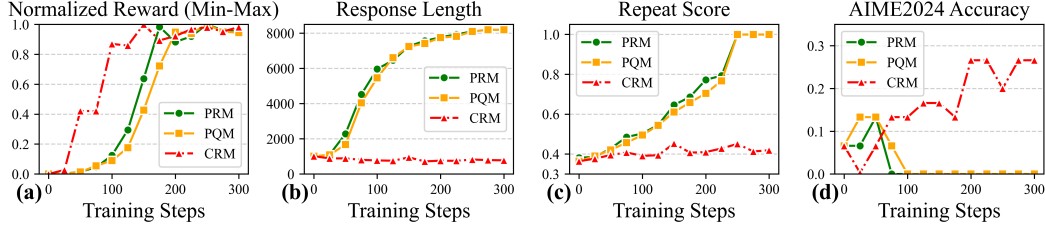

Figure 4: Evolution of (a) Normalized Reward (Min-Max), (b) Response Length, (c) Repeat Score, and (d) downstream task performance with training steps.

wards through excessively long, repetitive outputs, as reflected in their near-saturated repeat scores. This indicates that the reward fails to discriminate between genuine reasoning step and superficial repetition. In contrast, CRM tightly couples intermediate rewards to the final outcome, ensuring that reward faithfully reflects reasoning quality (**RQ1**). CRM's precise credit assignment stabilizes optimization and suppresses degenerate strategies, thereby enhancing robustness to reward hacking.

### 4.5.2 SELF-REFLECTION DURING REASONING PROCESS

Previous works (He et al., 2024; Liu et al., 2025b; Bensal et al., 2025) have observed that improvements in reasoning ability are often associated with a key phenomenon of self-reflection, in which models actively review and check previous steps. This phenomenon provides important insights into understanding and enhancing reasoning in LLMs. Following prior research (Yeo et al., 2025; Liu et al., 2025a;b), we define the self-reflection score as the aver-

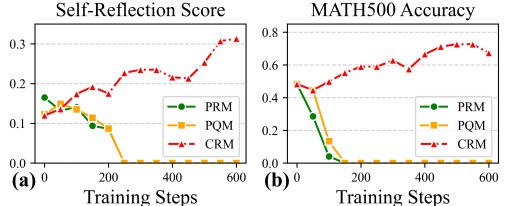

Figure 5: Evolution of self-reflection and downstream accuracy during RL training

age frequency of reflective expressions (e.g., "rethink", "let's check") appearing in a model's response, normalized by the output length (per 1000 tokens), which measures self-reflection capability during reasoning. The full list of reflective expressions is provided in the Appendix D.

**Finding 2: CRM encourages more self-reflection behaviors (RQ2).** As shown in Figure 5, the self-reflection score of CRM steadily rises during RL training, accompanied by improvements in downstream MATH500 accuracy, indicating that the model becomes increasingly reflective. In contrast, PRM and PQM show little to no growth in self-reflection and their MATH500 accuracy collapses early. The temporal co-movement between rising self-reflection and improving MATH500 accuracy suggests that CRM 's precise credit assignment fosters more meaningful reasoning behaviors, even without verifiable rewards.

### 4.5.3 ABLATION STUDY

**Ablation on loss for CRM training.** Our modeling approach employs three losses (Eq. 11, Eq. 12, Eq. 13). Among them, $\mathcal{L}_S$ and $\mathcal{L}_W$ are supervised by the label $l \in \{0, 1\}$ indicating whether the final answer is correct, with a one-to-one correspondence to samples. Removing these losses would reduce the number of training samples and undermine fair comparison. Therefore, we preserve $\mathcal{L}_S$ and $\mathcal{L}_W$ while conducting ablations on $\mathcal{L}_z$. The

Table 4: Ablation results for $\mathcal{L}_z$.

| Proportion of data used for $\mathcal{L}_z$ | @8 | @16 | @32 | @64 | @128 |
|---|---|---|---|---|---|
| 0% | 47.0 | 44.2 | 41.6 | 39.2 | 38.2 |
| 10% | 52.4 | 51.2 | 50.6 | 49.6 | 47.6 |
| 25% | 54.0 | 52.4 | 53.6 | 53.2 | 52.0 |
| 50% | 54.4 | 53.6 | 57.2 | 55.8 | 55.0 |
| 100% | 53.0 | 56.4 | 56.6 | 55.8 | 56.6 |

loss $\mathcal{L}_z$ directly encourages the model to identify the exact step at which a wrong state occurs. Specifically, we set the total amount of data in the dataset applicable to $\mathcal{L}_z$ as 100% and apply scaling, evaluating CRM with 0%, 10%, 25%, and 50% of the data under the Best-of-N sampling on the MATH500 dataset using Qwen2.5-3B-Instruct.

**Finding 3: CRM exhibits high data efficiency (RQ3).** The ablation results in Table 4 show that even a very small proportion of data applicable to $\mathcal{L}_z$ leads to substantial performance gains: moving from 0% to just 10% supervision produces a large improvement in Best-of-N accuracy. Increasing the proportion of data yields additional benefits, but the improvements quickly saturate, with 50% already achieving near-optimal performance. This demonstrates that CRM requires only limited data for $\mathcal{L}_z$ to achieve strong results, highlighting its high data utilization efficiency. A more detailed analysis of the reasons behind this efficiency, as well as comparisons with the baseline, can be found in the Appendix J.

### 4.5.4 Validation on More Domains

To further validate the effectiveness of our CRM, we extend our evaluation to additional domains beyond mathematics. We leverage the multi-domain dataset MMLU-Pro-CoT-Train from VersaPRM (Zeng et al., 2025a) as it provides reasoning processes with step-wise labels across a wide range of domains. We train the reward models including CRM and the baselines on this dataset using Qwen2.5-3B-Instruct as backbone and subsequently assess their performance using Best-of-N sampling on five distinct domains including biology, business, health, history, and physics.

**Finding 4: CRM can be extended to other domains and demonstrates strong performance. (RQ4).** As shown in Table 5, CRM outperforms the baselines in almost all tested domains. This demonstrates that CRM's validity is not limited to mathematical tasks and can be applied to a broader range of domains.

Table 5: Best-of-N accuracy comparison across different domains (MMLU-Pro-CoT-Eval).

| Methods | Biology | | | Business | | | Health | | | History | | | Physics | | |
|---|---|---|---|---|---|---|---|---|---|---|---|---|---|---|---|
| | @16 | @32 | @64 | @16 | @32 | @64 | @16 | @32 | @64 | @16 | @32 | @64 | @16 | @32 | @64 |
| PRM | 80.0 | 79.2 | 79.2 | 64.8 | 64.8 | 66.2 | 60.7 | 57.1 | 57.1 | 48.0 | 48.7 | 48.7 | 61.0 | 58.9 | 60.3 |
| PQM | 80.8 | 80.0 | 80.8 | 64.1 | 63.5 | 65.5 | 62.1 | 61.4 | 61.4 | 50.0 | 51.3 | 49.3 | 61.6 | 58.2 | 62.3 |
| **CRM (ours)** | 83.1 | 83.1 | 82.3 | 66.2 | 66.9 | 68.2 | 61.4 | 62.1 | 60.7 | 50.7 | 52.0 | 51.3 | 64.4 | 63.7 | 66.4 |

## 5 Conclusion

In this paper, we introduce CRM, which frames LLM reasoning as a temporal probabilistic process. By explicitly modeling the causal dependencies between steps and linking process to the outcome, CRM addresses the limitations of isolated step modeling and limited outcome awareness in prior work. Experiments across Best-of-N sampling, beam search, and RL demonstrate that CRM consistently outperforms strong baselines, improves cross-sample comparability, and is robust to reward hacking. Building on these results, we plan to extend CRM to broader domains and task formats, and we hope this work catalyzes further research on reward-model–driven RL for reasoning, moving beyond reliance on ground truth to achieve broader generalization.

## 6 Acknowledgments

The research received support from National Natural Science Foundation of China (Grant No. 62406193). This work was also supported by ShanghaiTech AI Initiative (Grant No. AI2026B08). The authors gratefully acknowledge further assistance provided by Shanghai Frontiers Science Center of Human-centered Artificial Intelligence, MoE Key Lab of Intelligent Perception and Human-Machine Collaboration, and HPC Platform of ShanghaiTech University.

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

# A  IMPLEMENTATION DETAILS

All experiments are conducted on 8 NVIDIA H20 GPUs.

## A.1  REWARD MODEL TRAINING

For data processing, we follow prior work (Yu et al., 2024; Zheng et al., 2025; Zhang et al., 2025b), using newline characters to indicate boundaries between individual reasoning steps. In terms of implementation, we used the ZeRO-2 optimization stage of DeepSpeed with bfloat16 precision to train the model. We employed AdamW optimizer with a learning rate of 5e-6 and a batch size of 32.

## A.2  BEST-OF-N SAMPLING

For the Best-of-N sampling evaluation, we adopt Llama-3.1-8B-Instruct as the generator and generate trajectories using the vLLM pipeline with temperature=1, top-p=1, and a maximum length of 2048. Each generated trajectory is subsequently scored by the reward models. Consistent with the settings in their original papers and code implementations, PRM, PQM, and IPRM aggregate process rewards into a sequence-level score by taking the minimum value across all steps. In contrast, our CRM computes $S(T)$ as the trajectory-level score.

## A.3  BEAM SEARCH

Our beam search is configured with four total sampling numbers, denoted as $N \in \{4, 8, 20, 100\}$. For each value of $N$, we test three distinct beam sizes, $b$. Initially, $N$ candidate responses are generated from a given question. In each subsequent step, the reward model evaluates the current set of candidates and selects the top-scoring $b$ trajectories as prefixes for expansion, where each prefix produces $N/b$ new continuations. This iterative process continues until all trajectories have generated a complete answer or the predefined maximum step count or token length is reached. The trajectory with the highest reward is then selected to extract the final answer.

For the evaluation of the Qwen2.5-Math-1.5B and Qwen2.5-Math-7B reward models, we adopt Qwen2.5-3B as the generator, and we adopt Llama3.1-8B as the generator for its corresponding reward model. All rollout generation is performed using the vLLM pipeline, with a temperature of 0.7 and top-p of 1. We set a maximum token length of 4096 and a maximum of 30 steps for each trajectory generation.

## A.4  RL OPTIMIZATION

**Advantage Estimation.**  The reward model provides step-wise process rewards $r$, and we adopt RLOO (Leave-One-Out) (Ahmadian et al., 2024) to estimate the advantage based on these rewards. RLOO uses multiple model outputs other than the current one to compute a baseline, effectively reducing variance in advantage estimation. Since RLOO is originally defined at the sequence level, following prior work (Cheng et al., 2025), we employ its token-level variant. Specifically, given a question, the LLM generates $K$ responses $\{y_i\}_{i=1}^K$, each containing at most $M$ tokens. Let $r_i^t$ denote the reward of the $t$-th token in the $i$-th response, and $A_i^t$ denote its corresponding advantage. In our implementation, we set the discount factor $\gamma$ to 1. The advantage is computed as:

$$A_i^t = \sum_{j=t}^M r_i^j \; - \; \frac{\sum_{k \neq i} \sum_{l=1}^M \sum_{j=l}^M r_k^j}{(K-1)M} \tag{15}$$

**Policy Update.**  We optimize the policy model $\pi_\theta$ by maximizing the following objective:

$$L(\theta) = \mathbb{E}_{i,t} \left[ \min\left( \frac{\pi_\theta(y_i^t \mid y_i^{<t})}{\pi_{\theta_{old}}(y_i^t \mid y_i^{<t})} A_i^t, \; \text{clip}\left( \frac{\pi_\theta(y_i^t \mid y_i^{<t})}{\pi_{\theta_{old}}(y_i^t \mid y_i^{<t})}, 1 - \epsilon, \; 1 + \epsilon \right) A_i^t \right) - \beta \mathbb{D}_{KL}(\pi_\theta || \pi_{ref}) \right] \tag{16}$$

where $\pi_{\theta_{old}}$ is the old policy and $\pi_{\theta_{ref}}$ is the reference model. $\epsilon$ is the PPO clipping threshold and $\beta$ controls the strength of the KL penalty; both are hyperparameters.

**Experimental setting.** We use Orz-Math-57k (Hu et al., 2025) as the RL training set. The dataset was created through a thorough data-cleaning process, ensuring no overlap with commonly used benchmarks. Experiments are conducted with veRL (Sheng et al., 2024) framework. We segment each LLM-generated response into steps using double line breaks and assign scores to each step with the reward model. We train with a fixed learning rate of 1e-6, a prompt batch size of 64, and 4 responses sampled per prompt, using a KL coefficient $\beta = 1e\text{-}3$ and a clip ratio $\epsilon = 0.2$. For generation, we adopt vLLM with temperature = 1, top-p=1, and a maximum length of 8192 tokens, while during testing we set temperature =0 and top-p=1.

**CRM+VR setting.** When used CRM alone, it provides step-wise rewards ($r_t^C$) for all steps, including the final reasoning step $T$. When combined with VR, only the final step additionally receives a verifiable reward ($r_T^V$), while earlier steps still rely solely on CRM. Formally, the reward at each step $t$ is defined as:

$$r_t = \begin{cases} r_t^C, & \text{if } t < T \\ r_T^C + r_T^V, & \text{if } t = T \end{cases} \tag{17}$$

## B  Full Experimental Results for Beam Search

The detailed specification of the beam sizes $b$ used for each $N$, along with the complete results, is provided in Table 6, Table 7 and Table 8.

Table 6: Beam Search performance of Qwen2.5-Math-1.5B on MATH500 and Gaokao2023

| $N$ | $b$ | MATH500 | | | | | GAOKAO2023 | | | | |
|---|---|---|---|---|---|---|---|---|---|---|---|
| | | ORM | PRM | PQM | IPRM | CRM | ORM | PRM | PQM | IPRM | CRM |
| 4 | 4 | 49.20 ± 0.60 | 49.93 ± 1.07 | 50.60 ± 0.80 | 44.27 ± 0.73 | 50.40 ± 0.20 | 34.46 ± 0.35 | 34.29 ± 1.30 | 35.06 ± 1.30 | 32.55 ± 1.47 | 34.03 ± 2.34 |
| | 2 | 50.73 ± 1.47 | 51.80 ± 0.80 | 52.60 ± 0.60 | 44.13 ± 1.27 | 53.27 ± 1.93 | 35.58 ± 1.82 | 34.55 ± 1.82 | 34.55 ± 1.56 | 31.00 ± 0.43 | 38.70 ± 0.26 |
| | 1 | 49.73 ± 2.27 | 50.27 ± 3.33 | 52.67 ± 0.53 | 43.67 ± 2.33 | 54.07 ± 2.73 | 32.99 ± 2.08 | 34.72 ± 1.39 | 36.88 ± 0.78 | 29.96 ± 1.21 | 36.71 ± 0.95 |
| 8 | 8 | 51.27 ± 0.33 | 52.53 ± 1.07 | 54.60 ± 2.20 | 47.27 ± 0.93 | 53.67 ± 1.53 | 38.18 ± 1.30 | 35.15 ± 1.73 | 37.58 ± 1.65 | 34.46 ± 0.87 | 36.54 ± 2.42 |
| | 4 | 54.80 ± 1.40 | 55.73 ± 1.67 | 56.20 ± 0.60 | 46.07 ± 1.33 | 57.47 ± 0.13 | 37.49 ± 1.47 | 37.84 ± 1.39 | 37.06 ± 0.35 | 32.38 ± 0.35 | 39.31 ± 1.47 |
| | 2 | 52.80 ± 0.60 | 54.13 ± 0.47 | 56.60 ± 0.80 | 45.27 ± 1.53 | 58.40 ± 1.30 | 36.28 ± 0.35 | 36.54 ± 1.90 | 38.61 ± 0.61 | 30.56 ± 0.35 | 39.74 ± 0.52 |
| 20 | 20 | 53.93 ± 0.67 | 55.73 ± 0.67 | 57.00 ± 0.20 | 47.73 ± 1.47 | 55.40 ± 1.40 | 37.75 ± 0.43 | 37.75 ± 0.43 | 39.05 ± 0.69 | 34.29 ± 1.30 | 36.62 ± 2.34 |
| | 10 | 56.80 ± 0.60 | 56.87 ± 0.53 | 58.87 ± 0.73 | 48.33 ± 1.87 | 59.80 ± 0.20 | 38.44 ± 1.04 | 38.53 ± 1.21 | 39.74 ± 2.08 | 35.32 ± 0.52 | 41.04 ± 0.78 |
| | 5 | 55.00 ± 0.40 | 55.80 ± 1.40 | 57.47 ± 1.53 | 47.07 ± 0.53 | 61.00 ± 0.60 | 37.66 ± 0.52 | 38.70 ± 0.52 | 40.61 ± 1.73 | 31.69 ± 0.78 | 40.78 ± 0.26 |
| 100 | 50 | 57.00 ± 1.00 | 57.80 ± 0.60 | 58.80 ± 0.60 | 46.67 ± 1.33 | 63.00 ± 0.40 | 40.17 ± 0.87 | 38.53 ± 0.95 | 39.39 ± 1.13 | 34.55 ± 1.30 | 39.91 ± 1.13 |
| | 25 | 58.07 ± 0.53 | 57.40 ± 0.40 | 57.73 ± 0.27 | 47.47 ± 1.13 | 61.47 ± 0.93 | 38.87 ± 0.87 | 38.96 ± 1.30 | 39.83 ± 1.73 | 32.99 ± 1.56 | 43.55 ± 0.61 |
| | 10 | 55.87 ± 0.33 | 58.00 ± 1.20 | 57.67 ± 1.53 | 46.13 ± 0.27 | 61.87 ± 0.73 | 38.70 ± 0.26 | 38.27 ± 0.95 | 39.05 ± 0.43 | 32.99 ± 0.26 | 42.08 ± 1.04 |

Table 7: Beam Search performance of Qwen2.5-Math-7B on MATH500 and Gaokao2023

| $N$ | $b$ | MATH500 | | | | | GAOKAO2023 | | | | |
|---|---|---|---|---|---|---|---|---|---|---|---|
| | | ORM | PRM | PQM | IPRM | CRM | ORM | PRM | PQM | IPRM | CRM |
| 4 | 4 | 51.47 ± 1.53 | 51.20 ± 1.40 | 50.87 ± 0.73 | 49.53 ± 1.07 | 51.20 ± 1.00 | 36.88 ± 0.52 | 34.55 ± 0.52 | 36.45 ± 1.73 | 35.67 ± 1.47 | 34.72 ± 0.35 |
| | 2 | 51.87 ± 0.73 | 52.13 ± 1.47 | 52.73 ± 0.67 | 45.73 ± 1.67 | 54.20 ± 0.60 | 37.49 ± 2.25 | 35.32 ± 0.26 | 37.84 ± 0.61 | 31.95 ± 1.56 | 38.53 ± 1.47 |
| | 1 | 51.53 ± 1.87 | 50.93 ± 0.47 | 52.27 ± 1.53 | 42.13 ± 0.87 | 56.07 ± 0.53 | 35.76 ± 1.13 | 37.58 ± 1.90 | 37.40 ± 0.52 | 32.29 ± 0.95 | 39.83 ± 1.73 |
| 8 | 8 | 54.60 ± 0.40 | 53.60 ± 0.60 | 56.53 ± 1.67 | 54.07 ± 0.13 | 56.00 ± 0.40 | 38.27 ± 1.21 | 37.32 ± 1.39 | 37.58 ± 0.87 | 38.53 ± 1.99 | 38.87 ± 0.35 |
| | 4 | 57.67 ± 1.33 | 55.67 ± 0.13 | 57.87 ± 1.53 | 50.27 ± 0.33 | 60.60 ± 1.40 | 40.26 ± 0.26 | 40.52 ± 0.78 | 39.65 ± 1.39 | 34.03 ± 1.04 | 42.60 ± 1.04 |
| | 2 | 56.27 ± 0.93 | 54.07 ± 1.13 | 56.07 ± 0.53 | 46.33 ± 1.07 | 58.27 ± 2.33 | 39.57 ± 1.47 | 39.13 ± 1.90 | 40.61 ± 0.69 | 31.69 ± 0.52 | 42.77 ± 1.13 |
| 20 | 20 | 56.27 ± 1.13 | 56.73 ± 0.27 | 56.93 ± 1.47 | 54.20 ± 2.40 | 57.80 ± 1.40 | 41.30 ± 1.30 | 41.04 ± 0.52 | 39.83 ± 0.69 | 40.26 ± 1.04 | 41.90 ± 0.69 |
| | 10 | 59.47 ± 1.53 | 59.93 ± 0.87 | 59.07 ± 0.93 | 52.73 ± 1.67 | 61.40 ± 1.20 | 44.07 ± 0.61 | 40.78 ± 0.52 | 41.21 ± 0.61 | 37.14 ± 1.56 | 45.37 ± 0.61 |
| | 5 | 58.80 ± 0.80 | 57.07 ± 0.53 | 59.20 ± 1.20 | 49.47 ± 1.13 | 62.87 ± 0.53 | 42.16 ± 2.51 | 40.61 ± 2.25 | 42.60 ± 0.52 | 35.41 ± 1.21 | 46.49 ± 0.52 |
| 100 | 50 | 59.67 ± 1.73 | 58.40 ± 0.80 | 59.07 ± 0.53 | 52.60 ± 0.80 | 63.80 ± 1.00 | 43.55 ± 0.61 | 43.12 ± 1.30 | 42.68 ± 0.69 | 39.57 ± 2.51 | 48.31 ± 2.60 |
| | 25 | 60.73 ± 0.67 | 60.13 ± 0.67 | 61.13 ± 0.87 | 51.73 ± 0.47 | 63.60 ± 1.00 | 43.64 ± 2.60 | 43.81 ± 0.35 | 43.03 ± 1.13 | 37.49 ± 0.95 | 47.62 ± 1.21 |
| | 10 | 59.47 ± 0.53 | 59.27 ± 1.13 | 60.20 ± 1.80 | 48.80 ± 1.20 | 64.07 ± 0.33 | 43.72 ± 1.73 | 41.90 ± 0.43 | 43.29 ± 0.87 | 34.89 ± 0.95 | 48.40 ± 1.21 |

## C  Reward Hacking

### C.1  Repeat Score

To quantify the degree of redundancy in generated responses, we define the *repeat score* based on $n$-gram statistics. This metric is a widely used indicator of textual repetition in the literature. Given a text sequence, we first normalize it by lowercasing, unifying quotation marks, and removing code blocks, LaTeX environments, and inline formulas. The text is then tokenized into alphanumeric tokens and basic punctuation.

Table 8: Beam Search performance of Llama3.1-8B on MATH500 and Gaokao2023

| N | b | MATH500 | | | | | GAOKAO2023 | | | | |
|---|---|---|---|---|---|---|---|---|---|---|---|
| | | ORM | PRM | PQM | IPRM | CRM | ORM | PRM | PQM | IPRM | CRM |
| 4 | 4 | $37.60 \pm 1.00$ | $37.67 \pm 0.33$ | $38.27 \pm 0.93$ | $37.07 \pm 0.13$ | $40.20 \pm 0.40$ | $25.37 \pm 0.61$ | $26.84 \pm 1.47$ | $25.37 \pm 0.35$ | $26.49 \pm 1.56$ | $26.93 \pm 2.16$ |
| | 2 | $38.13 \pm 0.47$ | $37.67 \pm 1.53$ | $39.20 \pm 0.80$ | $34.27 \pm 1.33$ | $39.60 \pm 0.80$ | $25.63 \pm 1.13$ | $26.41 \pm 0.87$ | $26.58 \pm 0.43$ | $23.72 \pm 1.73$ | $25.89 \pm 1.90$ |
| | 1 | $36.67 \pm 1.53$ | $37.87 \pm 0.93$ | $38.20 \pm 0.60$ | $32.73 \pm 0.87$ | $37.47 \pm 0.93$ | $23.55 \pm 0.87$ | $26.23 \pm 1.82$ | $25.54 \pm 1.47$ | $22.34 \pm 0.52$ | $25.80 \pm 0.69$ |
| 8 | 8 | $38.80 \pm 1.80$ | $39.07 \pm 0.73$ | $39.47 \pm 1.93$ | $38.67 \pm 1.73$ | $38.67 \pm 0.53$ | $25.45 \pm 1.30$ | $26.93 \pm 0.35$ | $27.71 \pm 0.61$ | $25.89 \pm 2.42$ | $28.40 \pm 0.43$ |
| | 4 | $38.73 \pm 1.87$ | $39.07 \pm 0.33$ | $40.47 \pm 1.13$ | $36.73 \pm 1.27$ | $41.00 \pm 1.40$ | $26.93 \pm 0.87$ | $28.23 \pm 0.35$ | $26.93 \pm 3.20$ | $23.98 \pm 2.25$ | $26.84 \pm 0.43$ |
| | 2 | $38.07 \pm 0.53$ | $39.67 \pm 1.33$ | $40.27 \pm 0.73$ | $34.40 \pm 0.60$ | $39.20 \pm 0.80$ | $24.07 \pm 2.94$ | $28.66 \pm 1.99$ | $26.32 \pm 0.69$ | $23.46 \pm 0.95$ | $26.15 \pm 0.87$ |
| 20 | 20 | $36.33 \pm 1.07$ | $40.13 \pm 1.87$ | $41.00 \pm 2.80$ | $37.13 \pm 0.87$ | $40.53 \pm 0.67$ | $26.75 \pm 0.52$ | $27.27 \pm 1.04$ | $28.31 \pm 1.30$ | $26.15 \pm 2.16$ | $27.62 \pm 0.69$ |
| | 10 | $38.93 \pm 1.07$ | $39.80 \pm 2.40$ | $40.67 \pm 0.73$ | $34.07 \pm 1.33$ | $42.07 \pm 1.13$ | $29.35 \pm 0.52$ | $27.53 \pm 1.82$ | $26.75 \pm 1.30$ | $24.68 \pm 2.08$ | $28.31 \pm 1.04$ |
| | 5 | $37.73 \pm 1.47$ | $39.80 \pm 1.40$ | $40.40 \pm 1.20$ | $33.93 \pm 1.07$ | $40.80 \pm 0.40$ | $26.23 \pm 0.78$ | $27.97 \pm 1.65$ | $27.36 \pm 0.95$ | $25.02 \pm 4.33$ | $28.74 \pm 0.35$ |
| 100 | 50 | $36.67 \pm 0.73$ | $39.07 \pm 1.53$ | $41.27 \pm 1.53$ | $32.20 \pm 1.20$ | $40.67 \pm 0.33$ | $27.62 \pm 1.73$ | $27.79 \pm 0.78$ | $28.23 \pm 0.09$ | $23.46 \pm 0.17$ | $28.31 \pm 2.08$ |
| | 25 | $36.67 \pm 1.73$ | $39.53 \pm 0.67$ | $40.27 \pm 0.33$ | $33.13 \pm 1.07$ | $40.20 \pm 0.60$ | $27.19 \pm 1.39$ | $27.97 \pm 1.13$ | $27.53 \pm 0.78$ | $24.42 \pm 0.52$ | $29.96 \pm 0.69$ |
| | 10 | $35.33 \pm 0.87$ | $38.47 \pm 0.33$ | $38.47 \pm 0.33$ | $34.13 \pm 1.67$ | $41.00 \pm 1.40$ | $27.27 \pm 1.82$ | $27.97 \pm 0.87$ | $27.01 \pm 1.04$ | $22.94 \pm 1.21$ | $28.83 \pm 1.56$ |

For an $n$-gram size $n$, we compute the *Distinct-n* ratio (Li et al., 2016):

$$\text{Distinct-}n = \frac{|\{\text{unique } n\text{-grams}\}|}{\text{total } n\text{-grams}}.$$

The *Repetition-n* score is defined as:

$$\text{Repetition-}n = 1 - \text{Distinct-}n,$$

which ranges from 0 to 1, with larger values indicating a higher proportion of repeated $n$-grams.

In practice, since $n = 2$ and $n = 3$ are widely adopted choices in NLP research and have been shown to be effective for capturing local redundancy patterns, we follow this design in our implementation and compute *Repetition-2* and *Repetition-3*, taking their maximum as the final *repeat score*:

$$\text{Repeat Score} = \max(\text{Repetition-2}, \text{Repetition-3}).$$

## C.2 EXAMPLES OF REWARD HACKING

### Example 1

**Question:**
Given that the sum of the first $n$ terms of an arithmetic sequence $\{a_n\}$ is $S_n$, if $a_4 + a_6 = 18$, then $S_9$ equals ( ).
A: $99$   B: $81$   C: $72$   D: $63$

**LLM Response:**
To solve this problem, we need to use the properties of an arithmetic sequence. Let's break down the problem step-by-step.
1. Arithmetic Sequence Formula: The $n$-th term of an arithmetic sequence can be written as:

$$a_n = a_1 + (n - 1)d$$

2. Sum of the First $n$ Terms:

$$S_n = \frac{n}{2}(2a_1 + (n - 1)d)$$

3. Given Condition: We are given that $a_4 + a_6 = 18$. Using the formula:

$$a_4 = a_1 + 3d, \quad a_6 = a_1 + 5d$$

So,

$$a_4 + a_6 = 2a_1 + 8d = 18 \implies a_1 + 4d = 9$$

4. Sum of the First 9 Terms:

$$S_9 = \frac{9}{2}(2a_1 + 8d) = \frac{9}{2}(2a_1 + 8d) = 9 \cdot 99$$

Let's solve this problem step-by-step.
1. Arithmetic Sequence Formula:

2. Sum of the first 9 terms:    ⟵ Start generating repeated content.
3. Sum of the first 9 terms:
4. Sum of the first 9 terms:

...

Let's solve the first 9 terms:
Let's solve the first 9 terms:
Let's solve the first 9 terms:

...

Let
Let9
Let9
Let
Let
Let9
Let9

...

## Example 2

**Question:**

Among the 900 residents of Aimeville, there are 195 who own a diamond ring, 367 who own a set of golf clubs, and 562 who own a garden spade. In addition, each of the 900 residents owns a bag of candy hearts. There are 437 residents who own exactly two of these things, and 234 residents who own exactly three of these things. Find the number of residents of Aimeville who own all four of these things.

**LLM Response:**

To solve this problem, we can use the principle of inclusion and exclusion. Let's define the following sets:
- $D$ = the set of residents who own a diamond ring.
- $G$ = the set of residents who own a set of golf clubs.
- $S$ = the set of residents who own a garden spade.
- $C$ = the set of residents who own a bag of candy hearts (which is the entire population of Aimeville).
We are given the following information:
- $|D| = 195$
- $|G| = 367$
- $|S| = 562$
- $|C| = 900$
The number of residents who own exactly two of these things is 437.
The number of residents who own exactly three of these things is 234.
We need to find the number of residents who own all four things, i.e., $|D \cap G \cap S \cap C|$.
First, let's denote the number of residents who own exactly $k$ of these things by $N_k$. We know that:

$$N_2 = 437$$
$$N_3 = 234$$

...

Find the values of f(x) in [-pi / 4]    ⟵ Start generating repeated content.
Find the values of f(x) in [-pi / 4]
Find the values of f(x) in [-pi / 4]

...

Find the maximum and minimum values of f(x) in [-pi / 4]
Find the maximum and minimum values of f(x) in [-pi / 4]
Find the maximum and minimum values of f(x) in [-pi / 4]
Find the maximum and minimum values of f(x) in [-pi / 4]

```
...
The values of f(x) in
The values of f(x) in
The values of f(x) in
The values of f(x) in
...
```

## D    REFLECTIVE EXPRESSIONS

Table 9 lists the reflective expressions used to compute the self-reflection score.

Table 9: Reflective expressions used in the computation of the self-reflection score.

| Reflective Expressions | Example |
| --- | --- |
| wait | "wait, let me think" |
| recheck | "recheck this step" |
| retry | "retry the calculation" |
| try again | "let's try again" |
| alternatively | "alternatively, we can ..." |
| however | "however, this may fail" |
| rethink | "rethink the argument" |
| let's check | "let's check the result" |
| let's verify | "let's verify the answer" |

## E    CROSS-SAMPLE COMPARISON CAPABILITY

**PQM focuses on intra-sample ranking but lacks cross-sample comparability.**    The core modeling principle of PQM (Li & Li, 2025) focuses on the relative ordering of steps within a sample, rather than modeling absolute values. For instance, applying the same shift operation to the $Q$-values of all steps within a sample still yields the same optimal ordering. As a result, the reward magnitudes across different samples do not share a consistent meaning. As stated in Corollary D.1 of the PQM paper, PQM allows comparison only when two trajectories share the same correct prefix. The limitation is validated by its suboptimal performance in our Best-of-N and beam search experiments (Section 4.2, Section 4.3), further demonstrating PQM's deficiency in cross-sample comparability.

**IPRM's lack of modeling for relationships between reasoning steps leads to poor cross-sample comparability.**    A core deficiency of IPRM (Yuan et al., 2025) is that its training relies solely on the final outcome, which leads to ambiguous credit assignment, as the model only ensures that the aggregate of all step-level rewards aligns with the final result, without enforcing more rigorous constraints on the process. Consequently, it lacks explicit modeling of the fine-grained causal relationships between reasoning steps. The process reward for any given step is not necessarily coherent with the preceding trajectory, resulting in inconsistent process rewards that cannot be reliably compared across different samples.

## F    CASE STUDIES ON PROCESS REWARDS

Figure 6 presents two case studies that compare the scoring behavior of different reward models on reasoning trajectories from the MathShepherd validation set. Each case visualizes the predicted reward per step (top) and reward drop between consecutive steps (bottom). The background is shaded to distinguish correct (green) and incorrect (red) reasoning steps, with a dashed line marking the error occurrence.

**Our CRM precisely identifies reasoning errors.**    ① The ORM fails to provide meaningful process rewards. Its rewards are volatile and poorly aligned with the correctness of intermediate steps.

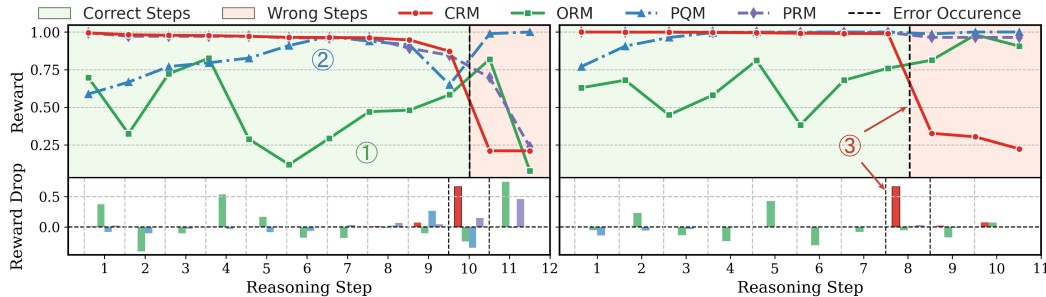

Figure 6: Case studies of different reward model scoring on trajectories from the MathShepherd validation set.

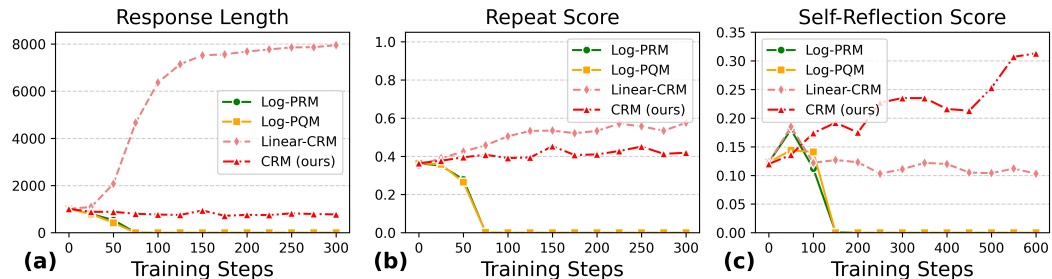

Figure 7: Training dynamics of ablation variants: (a) Response Length, (b) Repeat Score, and (c) Self-Reflection Score over training steps.

② The PQM, while capturing local ranking dependencies between reasoning steps, receives only indirect supervision from the final outcome. Consequently, it may incorrectly assign high rewards after a fatal error, failing to provide a precise assessment of the final result. ③ In contrast, our CRM overcomes these issues and precisely identifies which reasoning step enters an incorrect state. It maintains high rewards for correct steps, exhibits an immediate drop when the first error occurs, and subsequently maintains a low reward for all following incorrect steps.

## G ABLATION STUDY ON PROCESS REWARD FORMULATION OF CRM

Table 10: Pass@1 accuracy across benchmarks for ablation variants.

| Method | MATH500 | Minerva Math | OlympiadBench | AIME25 | AIME24 | AMC23 |
|---|---|---|---|---|---|---|
| Log-PRM | 71.4 | 33.0 | 32.2 | 13.3 | 16.6 | 45.0 |
| Log-PQM | 71.4 | 33.8 | 34.1 | 13.3 | 20.0 | 45.0 |
| Linear-CRM | 71.4 | 33.0 | 32.2 | 13.3 | 13.3 | 57.5 |
| CRM | **77.8** | **40.0** | **39.3** | **23.3** | **43.3** | **67.5** |

Our process reward is defined as $r = \log(1-h)$, derived from the theoretical analysis in Section 3.2. To validate the necessity of this formulation, we conduct experiments from two perspectives: (1) apply a log transform to baselines (PRM and PQM) score, i.e., redefining their original per-step reward $r_t^o$ as $r_t = \log r_t^o$, denoted as Log-PRM and Log-PQM; and (2) further test a linearized variant of CRM that replaces the original reward with $r = 1 - h$; we refer to this as Linear-CRM. Figure 7 shows that Log-PRM/Log-PQM and Linear-CRM quickly exhibit reward hacking where response length collapses and self-reflection vanishes, whereas CRM remains stable and steadily increases self-reflection. Table 10 mirrors this trend: CRM attains the best accuracy on every benchmark with clear margins over all ablations. These results underscore the necessity of

our theoretically grounded shaping $r = \log(1 - h)$, as it offers a stable and informative process reward that effectively mitigates reward hacking.

## H   COMPARISON WITH REASONING-ENHANCED PRMS

Recent work (Chen et al., 2025; Guo et al., 2025b; She et al., 2025; Zhao et al., 2025) integrates explicit reasoning capabilities into reward modeling. To compare our CRM with this line of work, we evaluate it against the representative reasoning-enhanced reward model R-PRM (She et al., 2025) and GenPRM (Zhao et al., 2025).

Table 11: Best-of-N accuracy comparison with reasoning-enhanced PRMs on MATH500.

| Method | Best-of-8 | Best-of-16 |
|---|---|---|
| R-PRM | 49.8 | 51.2 |
| GenPRM | 50.8 | 55.2 |
| **CRM (ours)** | **53.0** | **56.4** |

We evaluated these reward models using their official released checkpoints, the same settings and inference pipelines to score identical responses from our generator on the MATH500 dataset under Best-of-N sampling. The results are shown in Table 11. Compared to both baselines, CRM consistently achieves the highest performance. Specifically, under the Best-of-8 setting, CRM achieves 53.0, surpassing R-PRM (49.8) and GenPRM (50.8). Similarly, under the Best-of-16 setting, CRM reaches 56.4, outperforming both R-PRM (51.2) and GenPRM (55.2). These results highlight CRM's stronger capability in identifying high-quality responses compared to recent reasoning-enhanced and generative approaches.

## I   BEST-OF-N SAMPLING PERFORMANCE CHARACTERISTICS

In this section, we provide a more in-depth analysis of the performance characteristics of Best-of-N sampling. In BoN, an LLM generator first samples $N$ candidates for each query. Subsequently, a reward model selects the response with the highest predicted reward. Therefore, the LLM generator plays a role in shaping BoN performance. We observe that different LLM generators lead to distinct behaviors: BoN performance may continue to improve as $N$ increases, or it may converge at a small $N$ and show no further gains. As shown in Table 12, with a strong generator (Llama-3.1-8B-Instruct), all methods converge at small $N$ and do not improve further, whereas with a weaker generator (MetaMath-Mistral-7B), all methods exhibit increasing BoN performance as $N$ increases. This phenomenon can be explained as follows. (i) Weaker LLM generators have a lower probability of producing correct responses. At small $N$, they may generate no correct candidates, leaving the reward model with nothing correct to select. As $N$ increases, the sampled candidates are more likely to include correct answers, so the BoN performance starts to improve. (ii) Stronger LLM generators are able to produce correct responses even with small $N$. In this case, the BoN performance is not limited, leading to faster convergence.

These points are further supported by the Pass@N metric, which measures the probability that the LLM generator produces at least one correct response among the $N$ responses, serving as an upper bound on the achievable performance of BoN sampling with any reward model. Table 13 reports the Pass@N of the two LLM generators. The weaker generator MetaMath-Mistral-7B shows a substantial gap between Pass@8 and Pass@128, with a relatively low Pass@8. This indicates that when $N$ is small, the low upper bound limits BoN performance. The stronger LLM generator LLaMA-3.1-8B-Instruct achieves high Pass@N even for small $N$, leading to early convergence.

## J   DATA EFFICIENCY ANALYSIS OF CRM.

To further substantiate the data efficiency of CRM, we added PRM for comparison. We refer to the data from the PRM training set, excluding the final step, as process data. We conducted experiments using the same data proportion as in our ablation study, except for 0% (since PRM requires process data). As shown in Table 14, CRM reaches near-optimal performance with only 50% of the training data, whereas PRM still exhibits a noticeable performance gap under the same data budget.

Table 12: Best-of-N sampling performance across methods using different LLM generators.

| LLM Generator | Method | MATH500 | | | | |
| | | @8 | @16 | @32 | @64 | @128 |
|---|---|---|---|---|---|---|
| MetaMath-Mistral-7B | Pass@N | 53.0 | 62.0 | 69.6 | 75.8 | 80.4 |
| | PRM | 34.0 | 35.8 | 35.2 | 37.4 | 38.0 |
| | PQM | 36.4 | 38.4 | 37.8 | 38.6 | 40.4 |
| | **CRM (ours)** | 35.6 | 39.0 | 39.2 | 40.2 | 41.6 |
| LLaMA-3.1-8B-Instruct | Pass@N | 73.8 | 79.2 | 84.6 | 87.4 | 90.2 |
| | PRM | 54.2 | 55.2 | 55.2 | 54.2 | 54.6 |
| | PQM | 53.2 | 54.4 | 54.8 | 54.8 | 55.8 |
| | **CRM (ours)** | 53.0 | 56.4 | 56.6 | 55.8 | 56.6 |

Table 13: Pass@N of different LLM generators on MATH500 and GSM-Plus datasets.

| LLM Generator | MATH500 (Pass@N) | | | | | | GSM-Plus (Pass@N) | | | | | |
| | $\Delta[@8{\to}@128]$ | @8 | @16 | @32 | @64 | @128 | $\Delta[@8{\to}@128]$ | @8 | @16 | @32 | @64 | @128 |
|---|---|---|---|---|---|---|---|---|---|---|---|---|
| MetaMath-Mistral-7B | **27.4** | 53.0 | 62.0 | 69.6 | 75.8 | 80.4 | **11.1** | 73.2 | 78.1 | 80.5 | 83.0 | 84.3 |
| LLaMA-3.1-8B-Instruct | **16.4** | 73.8 | 79.2 | 84.6 | 87.4 | 90.2 | **6.3** | 79.2 | 81.2 | 82.2 | 83.3 | 85.5 |

The efficiency gain stems from how CRM leverages the supervision at step $t = z$. While PRM treats each step independently and learns a step-wise correctness classifier, CRM instead models reasoning as a temporal progression toward the final answer. By framing the entire reasoning trajectory as a sequence converging to correctness, as derived in Eqs. 11, 12, and 13, CRM allows the supervision signal at step $t = z$ to propagate backward to all earlier steps $t' \leq z$. This provides richer and more coherent training signals and enables CRM to make more effective use of available data, resulting in significantly improved data efficiency.

Table 14: Best-of-N accuracy on Math500 using varying proportions of data for CRM's $\mathcal{L}_z$ and PRM.

| Proportion of data used for $L_z$ | CRM | | | | | Process data used for PRM | PRM | | | | |
| | @8 | @16 | @32 | @64 | @128 | | @8 | @16 | @32 | @64 | @128 |
|---|---|---|---|---|---|---|---|---|---|---|---|
| 0% | 47.0 | 44.2 | 41.6 | 39.2 | 38.2 | 0% | - | - | - | - | - |
| 10% | 52.4 | 51.2 | 50.6 | 49.6 | 47.6 | 10% | 48.0 | 49.4 | 48.6 | 48.0 | 47.4 |
| 25% | 54.0 | 52.4 | 53.6 | 53.2 | 52.0 | 25% | 53.0 | 54.0 | 54.6 | 52.6 | 50.8 |
| 50% | 54.4 | 53.6 | 57.2 | 55.8 | 55.0 | 50% | 52.4 | 54.2 | 55.2 | 53.4 | 52.0 |
| 100% | 53.0 | 56.4 | 56.6 | 55.8 | 56.6 | 100% | 54.2 | 55.2 | 55.2 | 54.2 | 54.6 |

