# OpenReview forum: "Linking Process to Outcome: Conditional Reward Modeling for LLM Reasoning"
_ICLR.cc/2026/Conference — ICLR 2026 Poster_

### Official Review · Reviewer_1Jdv · 2025-10-15

**Soundness:** 3
**Presentation:** 3
**Contribution:** 3
**Rating:** 6
**Confidence:** 4

**Summary:**

The paper proposes a new theoretical framework with Potential-Based Reward Shaping to train PRMs.  Their framework not only captures the dependencies between reasoning steps, but also between the reasoning steps and the final outcome. The empirical studies also validate the effectiveness of their frameworks.

**Strengths:**

- The intuition behind the framework is well-motivated and easy to follow.
- The empirical studies are comprehensive, including BoN, PRM-guided beam-search, and RL experiments. The cross-sample comparability appears appealing.

**Weaknesses:**

> Ambiguity of why choose $\log S(t)$ as the potential function.

The authors directly claim "A natural choice for the potential function is one that estimates the likelihood of eventually reaching a correct answer from the current state", but abridge the motivations and insights behind this substitution. Since this substitution is the key step to derive the final objective, the ambiguity here confuses.

**Questions:**

> Implementation details on how to combine CRM + VR would enhance the reproducibility.

> Finding 3 (Line 466) claims CRM exhibits a high data efficiency, but without comparison with previous works. Does CRM exhibits a higher  or on-par data efficiency compared to previous works?

> Confused about what each peak of CRM in Figure 1 means? It cannot show your CRM can capture dependency between reasoning steps and outcome.

---

> ### Author Response · Authors · 2025-11-22
> **Author Response to Reviewer 1Jdv (1/2)**
>
> We sincerely thank the reviewer for the detailed and helpful feedback.
>
> > Weaknesses: Ambiguity of why choose $\log S(t)$ as the potential function.
> The authors directly claim "A natural choice for the potential function is one that estimates the likelihood of eventually reaching a correct answer from the current state", but abridge the motivations and insights behind this substitution. Since this substitution is the key step to derive the final objective, the ambiguity here confuses.
>
> We apologize for the ambiguity and offer the following clarification.
>
> (i) The potential function in PBRS is usually chosen using expert domain knowledge [1].
> Specifically, PBRS is effective when the potential function encodes goal-directed progress [2,3,4,5].
> In our formulation, we define the potential function as $\log S(t)$, which denotes the log-likelihood of eventually reaching a correct answer from the current state, thereby explicitly capturing the progress toward the final correct answer.
>
> (ii) Using the log-likelihood form of $\log S(t)$ provides two main benefits:
> - It makes the potential function more stable and numerically well-behaved, particularly when the underlying likelihoods are very small, thereby avoiding numerical underflow and ensuring that the shaped reward reflects incremental progress in a smoother manner.
> - It simplifies the computation of the reward shaping term $\Phi(s_{t+1}) - \Phi(s_t)$ in Eq. (10), making the implementation straightforward and efficient.
>
> Thank you for your comment. We have revised the original statement in Line 203 of the paper, where all new changes are marked in blue.
>
> > Q1: Implementation details on how to combine CRM + VR would enhance the reproducibility.
>
> We appreciate the reviewer's interest in the CRM + VR setup. We provide the implementation details below.
>
> When CRM is used alone, it provides step-wise rewards ($r_t^{C}$) for all steps, including the final reasoning step $T$. When combined with VR, only the final step additionally receives a verifiable reward ($r_T^{V}$), while earlier steps still rely solely on CRM. Formally, the reward at each step $t$ is defined as:
> $$
> r_t =
> \begin{cases}
> r_t^{C}, & \text{if } t < T \\\\
> r_T^{C} + r_T^{V}, & \text{if } t = T
> \end{cases}
> $$
>
> After obtaining step‑wise rewards ($r_t$), we perform advantage estimation using RLOO, followed by a policy update (described in Appendix A.4).
>
> We have added these implementation details in Appendix A.4 (Line 820) of the revised manuscript.
> Furthermore, as mentioned in Appendix A of our paper, we will release our code to facilitate reproducibility.
> Thank you for your comments.

---

> ### Author Response · Authors · 2025-11-22
> **Author Response to Reviewer 1Jdv (2/2)**
>
> > Q2: Finding 3 (Line 466) claims CRM exhibits a high data efficiency, but without comparison with previous works. Does CRM exhibits a higher or on-par data efficiency compared to previous works?
>
> Thank you for your valuable feedback.
>
> As suggested, we conducted an additional comparison of data efficiency between our CRM and the baseline Process Reward Model (PRM) used in our paper.
> The results are presented in Table 1 below.
> CRM reaches near-optimal performance using only 50% of the training data, whereas PRM still shows a noticeable performance gap under the same data budget.
> This demonstrates that CRM achieves higher data efficiency than PRM.
>
> The efficiency gain stems from how CRM leverages the supervision at step $t=z$.
> While PRM treats each step independently and learns a step-wise correctness classifier, CRM instead models reasoning as a temporal progression toward the final answer.
> By framing the entire reasoning trajectory as a sequence converging to correctness, as derived in Eqs. (11)-(13) in our paper, CRM allows the supervision signal at step $t=z$ to propagate backward to all earlier steps $t' \leq z$.
> This provides richer and more coherent training signals and enables CRM to make more effective use of the available data, resulting in significantly improved data efficiency.
>
> The detailed experimental settings and analysis have been added to Appendix J of the revised manuscript.
> Thank you once again for your suggestions.
>
> Table 1. Best-of-N accuracy on MATH500 under varying data usage for different methods.
> |BoN Accuracy| CRM | | | | | PRM | | | | |
> |:---|:---|:---|:---|:---|:---|:---|:---|:---|:---|:---|
> | **Proportion** | @8 | @16 | @32 | @64 | @128 | @8 | @16 | @32 | @64 | @128 |
> | 10% | 52.4 | 51.2 | 50.6 | 49.6 | 47.6 | 48.0 | 49.4 | 48.6 | 48.0 | 47.4 |
> | 25% | 54.0 | 52.4 | 53.6 | 53.2 | 52.0 | 53.0 | 54.0 | 54.6 | 52.6 | 50.8 |
> | 50% | 54.4 | 53.6 | 57.2 | 55.8 | 55.0 | 52.4 | 54.2 | 55.2 | 53.4 | 52.0 |
> | 100%| 53.0 | 56.4 | 56.6 | 55.8 | 56.6 | 54.2 | 55.2 | 55.2 | 54.2 | 54.6 |
>
> > Q3: Confused about what each peak of CRM in Figure 1 means? It cannot show your CRM can capture dependency between reasoning steps and outcome.
>
> Sorry for the confusion.
>
> * In Figure 1, *each peak* corresponds to the distribution $h(t)$ at reasoning step $t$. As defined in Eq. (4) of our paper, $h(t)$ represents the probability that the LLM enters an incorrect reasoning state at step $t$, conditioned on all previous steps $1, \ldots, (t-1)$ being correct. This is further analyzed in Eqs. (5) to (10) in the paper, where we derive how the CRM's process reward is constructed from $h(t)$. The final process reward used for downstream LLM reasoning is given in Eq. (10).
>
> * In Figure 1, the *red solid arrows* indicate how intermediate step rewards contribute to the final correctness via the conditional probability chain rule in Eq. (5). The red dashed arrows illustrate the causal dependencies between reasoning steps that CRM captures through its conditional probability modeling in Eq. (4). Together, these elements visualize how CRM models the sequential dependency structure between intermediate steps and the final reasoning outcome.
>
> To better illustrate our modeling approach, we have revised Figure 1 by updating the representation of $h(t)$, and added explanatory text in the updated manuscript to reduce potential confusion.
> Thank you again for highlighting this point.
>
> [1] Andrew Y Ng, Daishi Harada, and Stuart Russell. Policy invariance under reward transformations: Theory and application to reward shaping. In Proceedings of the Sixteenth International Conference on Machine Learning (ICML), 1999.
>
> [2] Chunlok Lo, Kevin Roice, Parham Mohammad Panahi, Scott M. Jordan, Adam White, Gabor Mihucz, Farzane Aminmansour, and Martha White. Goal-space planning with subgoal models. Journal of Machine Learning Research, 25(330):1–57, 2024.
>
> [3] Henrik Müller and Daniel Kudenko. Improving the effectiveness of potential-based reward shaping in reinforcement learning. arXiv preprint arXiv:2502.01307, 2025.
>
> [4] Paulina Stevia Nouwou Mindom, Amin Nikanjam, Foutse Khomh, and John Mullins. On assessing the safety of reinforcement learning algorithms using formal methods. In 2021 IEEE 21st International Conference on Software Quality, Reliability and Security (QRS), pp. 260–269. IEEE, 2021.
>
> [5] Eric Wiewiora, Garrison W Cottrell, and Charles Elkan. Principled methods for advising reinforcement learning agents. In Proceedings of the 20th international conference on machine learning (ICML), pp. 792–799, 2003.

---

### Official Review · Reviewer_8bhE · 2025-10-31

**Soundness:** 2
**Presentation:** 3
**Contribution:** 2
**Rating:** 2
**Confidence:** 4

**Summary:**

This paper introduces Conditional Reward Modeling (CRM) — a framework for enhancing the reasoning capability of large language models by modeling the reasoning trajectory as a temporal probabilistic process. CRM explicitly links intermediate reasoning steps to the final outcome by defining step-level rewards as conditional probabilities of maintaining correct reasoning. This design aims to resolve credit assignment ambiguity and improve robustness against reward hacking.
Empirical results across Best-of-N sampling, beam search, and reinforcement learning (RL) demonstrate that CRM outperforms baseline reward models (PRM, PQM, IPRM) in reasoning tasks such as GSM-Plus, MATH500, and AIME benchmarks. The method shows stable optimization and increased self-reflection behaviors in RL-trained models.

**Strengths:**

1.	Formal rigor — CRM introduces a clear probabilistic derivation linking reasoning steps to final outcomes, which elegantly resolves step-level credit ambiguity.
	2.	Empirical robustness — Demonstrates consistent performance gains across multiple reasoning benchmarks and settings (Best-of-N, beam search, RL).
	3.	Reward hacking analysis — The paper includes a thoughtful investigation into degenerate reward optimization and shows CRM’s improved stability.
	4.	Data efficiency — The ablation on L_z shows CRM achieves high data utilization, improving with limited supervision.

**Weaknesses:**

1.	Flawed motivation — The key claim that verifiable rewards are costly and difficult to scale is unconvincing and outdated.
	- Recent large-scale reasoning systems such as DeepSeek-R1 have demonstrated that verifiable rewards are both accurate and scalable, making CRM’s justification inconsistent with current research trends.
	- Moreover, DeepSeek-R1 explicitly abandoned PRM-style process rewards for their instability, undermining the central motivation of this paper.
	2.	Lack of comparison to verifiable-reward models —
	- Although CRM shows improvement over PRM-family models (PRM, PQM, IPRM), the experiments exclude comparisons with contemporary verifier-based RL systems like Qwen3-RL, DeepSeek-R1, or OpenAI o1.
	- Without these baselines, it’s difficult to assess whether CRM’s improvements are competitive in modern reasoning settings.
	3.	Limited scalability and applicability —
	- CRM is only tested on LLM-based reasoning (Best-of-N, beam search, RL) but not on large reasoning models (LRMs), where process reasoning involves extended multi-branch trajectories and hierarchical dependencies.
	- The proposed CRM-based search mechanism may also introduce additional computational cost compared to verifier-based approaches, which is not analyzed or mitigated.
	4.	Incremental novelty —
	- The use of conditional probability chains and potential-based reward shaping is mathematically elegant but conceptually derivative of standard PRM extensions (e.g., IPRM, PQM).
	- The claimed robustness advantage could likely be replicated by integrating verifiable signals with existing PRMs.

**Questions:**

1.	How does CRM compare quantitatively to DeepSeek-R1 or Qwen3-RL in reasoning benchmarks (e.g., AIME24, GAIA)?
	2.	What is the computational overhead introduced by CRM’s step-level conditional modeling during beam search or RL?
	3.	Could CRM be integrated with verifiable rewards rather than replacing them, to combine interpretability with scalability?
	4.	Have you attempted to apply CRM to large reasoning models or long-horizon tasks beyond math reasoning (e.g., proof or planning tasks)?

---

> ### Author Response · Authors · 2025-11-22
> **Author Response to Reviewer 8bhE (1/4)**
>
> Thank you for your constructive feedback! We appreciate your recognition of our strengths and we hope the following clarifications can dispel your doubts.
>
> > Weakness 1:
>     Flawed motivation — The key claim that verifiable rewards are costly and difficult to scale is unconvincing and outdated.
>     - Recent large-scale reasoning systems such as DeepSeek-R1 have demonstrated that verifiable rewards are both accurate and scalable, making CRM's justification inconsistent with current research trends.
>     - Moreover, DeepSeek-R1 explicitly abandoned PRM-style process rewards for their instability, undermining the central motivation of this paper.
>
>
> First, we would like to clarify that a verifiable reward is a deterministic correctness signal obtained by checking whether the model's output matches the ground truth.
> It can guide policy training via RL, but it also faces the following limitations:
> - (i) Verifiable rewards rely on ground-truth, which often requires costly human annotations and limits scalability.
> - (ii) Verifiable rewards are available only at the final step of reasoning [1,2], leading to sparse reward issues.
>
> Second, the motivation of our paper is to address the limitations of existing reward models.
> - (i) Existing PRMs face isolated step modeling, where each reasoning step is assessed independently, neglecting the intrinsic sequential dependencies of reasoning.
> - (ii) These methods lack explicit modeling of the final outcome, which may lead to biased process rewards.
> - (iii) These limitations make downstream models vulnerable to reward hacking and lead to suboptimal performance.
>
> In fact, our motivation and the latest work [3,4,5] on reward models are not in conflict with verifiable rewards. The two are orthogonal.
> Furthermore, combining reward models with verifiable rewards is feasible and can overcome the limitations of pure verifiable rewards, leading to more effective training.
> For example, our proposed reward model can be combined with verifiable rewards to further enhance performance (Table 3 in our paper).
>
> Regarding DeepSeek-R1's mention that PRM-style rewards are prone to reward hacking, our method mitigates this issue through our innovative design (see Section 4.5.1), offering insights for more reliable reward models.
>
> > Weakness 2:
>     Lack of comparison to verifiable-reward models —
>     - Although CRM shows improvement over PRM-family models (PRM, PQM, IPRM), the experiments exclude comparisons with contemporary verifier-based RL systems like Qwen3-RL, DeepSeek-R1, or OpenAI o1.
>     - Without these baselines, it's difficult to assess whether CRM's improvements are competitive in modern reasoning set
>
> Thank you for your suggestion.
>
> We acknowledge that comparing with verifiable reward methods is important.
> We added a verifier-based RL baseline using the same base model as our method and evaluated it on multiple mathematical reasoning benchmarks.
> The results have been included in the revised manuscript (Line 376) and are also shown in Table 1 below. All new changes in the revised manuscript are marked in blue.
> The results have illustrated that our method outperforms the verifier-based RL approach, demonstrating its effectiveness and superiority.
>
> Table 1: Pass@1 accuracy evaluated on mathematical reasoning benchmarks.
> | | MATH500 | Minerva Math | OlympiadBench | AIME25 | AIME24 | AMC23 |
> | :--- | :--- | :--- | :--- | :--- | :--- | :--- |
> | Verifier-based RL | 76.2 | 38.6 | 38.0 | 16.6 | 30.0 | 62.5 |
> | CRM | 77.8 | 40.0 | 39.3 | 23.3 | 43.3 | 67.5 |
>
> However, ultra–large-scale models like DeepSeek-R1 and OpenAI-o1 are currently beyond the scope of our comparison, as their vastly higher resource budgets make a direct evaluation infeasible.
> Direct API-based comparisons are also impractical due to the difficulty of aligning experimental settings.
> We plan to revisit these models in future work when our experimental conditions allow.

---

> ### Author Response · Authors · 2025-11-22
> **Author Response to Reviewer 8bhE (2/4)**
>
> > Weakness 3.1:
>     Limited scalability and applicability —
>     - CRM is only tested on LLM-based reasoning (Best-of-N, beam search, RL) but not on large reasoning models (LRMs), where process reasoning involves extended multi-branch trajectories and hierarchical dependencies.
>
> Thank you for raising concerns about scalability and applicability.
>
> We would like to clarify that, in our understanding, the concepts of LLM-based reasoning and large reasoning models (LRMs) are not in conflict [6,7].
> Our extensive experiments fall well within the scope of the LRM paradigm.
>
> In our understanding, extended multi-branch trajectories refer to search over multiple reasoning branches. In our work, our CRM is explicitly employed to guide such multi-branch exploration via beam search.
> The corresponding results are reported in Table 2 of the original paper, which clearly demonstrate the effectiveness of our method in such scenarios.
>
>
> > Weakness 3.2:
>     - The proposed CRM-based search mechanism may also introduce additional computational cost compared to verifier-based approaches, which is not analyzed or mitigated.
>
> Thank you for your feedback!
>
> We believe that the major concern stems from model-based reward and outcome verifiable reward.
> Our CRM and other process reward models assign step-wise rewards, which *cannot* be easily obtained from verifier-based methods that only assign the outcome reward and require ground-truth for verification.
>
> As for CRM-based search, we would like to clarify that reward models including our CRM and other process reward models can guide search.
> In contrast, verifier-based methods cannot, as they can only provide outcome rewards based on the ground truth.
> Making verifier-based methods applicable to search would require step-wise verification, which is costly and often infeasible.
>
> However, all above discussion is orthogonal to the main contribution of our work: we introduce CRM to link process rewards with outcome rewards, while the search procedure is merely an evaluation method rather than part of CRM itself.
>
> > Weakness 4.1:
>     Incremental novelty —
>     The use of conditional probability chains and potential-based reward shaping is mathematically elegant but conceptually derivative of standard PRM extensions (e.g., IPRM, PQM).
>
> Thank you for your comment.
>
> We would like to clarify that CRM is not conceptually derivative from standard PRMs. Instead, it's a fundamentally distinct formulation designed to address the limitations of existing methods.
> Standard PRMs (e.g., IPRM, PQM) often suffer from isolated step modeling and limited outcome awareness. In contrast, CRM addresses these limitations through three aspects:
> - *Modeling Paradigm*. CRM treats LLM reasoning as a temporal process toward the correct answer, modeling step-wise dependencies through conditional probabilities.
> - *Optimization Objective*. CRM aligns each step's reward with the final outcome, enabling precise credit assignment, an aspect overlooked by prior methods.
> - *Accurate and Robust Reward*. The consistent probabilistic modeling of CRM enables better cross-sample comparison, demonstrates effectiveness across multiple downstream tasks, and enhances robustness against reward hacking.
>
> > Weakness 4.2:
>     The claimed robustness advantage could likely be replicated by integrating verifiable signals with existing PRMs.
>
> Thank you for your insightful feedback!
>
> We would like to address your concern through the following three points:
>
> - (i) Our CRM falls within the scope of reward models and is robust to reward hacking without relying on external verifiable approaches.
>
> - (ii) Simply integrating verifiable signals with existing PRMs cannot replicate CRM. Existing PRMs often suffer from isolated step modeling and limited outcome awareness. Verifiable signals are sparse and cannot correct these limitations when integrated with existing PRMs.
>
> - (iii) Finally, the integration of verifiable signals is orthogonal to our method. Incorporating verifiable signals further enhances CRM's performance, as demonstrated in Section 4.4 (Table 3) of our paper.

---

> ### Author Response · Authors · 2025-11-22
> **Author Response to Reviewer 8bhE (3/4)**
>
> > Q1:
>     How does CRM compare quantitatively to DeepSeek-R1 or Qwen3-RL in reasoning benchmarks (e.g., AIME24, GAIA)?
>
> As detailed in our response to Weakness 2, a direct comparison with ultra-large-scale models like DeepSeek-R1 is infeasible due to limited computational resources and significant differences in model size. However, to fairly evaluate our method against verifiable reward methods, we added a verifier-based RL baseline using the identical backbone model. The results are shown in Table 1 above. Notably, on the AIME24 dataset, CRM achieves significantly better performance (43.3%) compared to the verifier-based RL baseline (30.0%), demonstrating the superiority of our method.
>
>
> > Q2:
>     What is the computational overhead introduced by CRM's step-level conditional modeling during beam search or RL?
>
> Thank you for your question.
>
> The computational overhead of our CRM is identical to that of existing PRM methods [8,9,10,11], with no additional overhead introduced when applied to beam search and RL.
>
> While step-level conditional modeling is our theoretical framework, the usage of CRM during inference remains the same as standard PRMs.
>
> > Q3:
>     Could CRM be integrated with verifiable rewards rather than replacing them, to combine interpretability with scalability?
>
> That's a great point and we thank you for your suggestion!
>
> As shown in Table 3 of our original paper, we found that CRM and verifiable rewards are indeed complementary rather than mutually exclusive, which aligns with your insight.
>
> However, regarding scalability, we would like to clarify that our CRM alone provides supervision for RL training without relying on ground truth, thereby enabling scalability. In contrast, integrating verifiable rewards reintroduces a dependency on ground truth labels, which limits the scalability.

---

> ### Author Response · Authors · 2025-11-22
> **Author Response to Reviewer 8bhE (4/4)**
>
> > Q4:
>     Have you attempted to apply CRM to large reasoning models or long-horizon tasks beyond math reasoning (e.g., proof or planning tasks)?
>
> Thank you for your suggestion.
>
> Directly applying CRM to ultra-large-scale LRMs requires immense effort and computational resources, while offering little insight to our work.
> We have evaluated our method and baselines on 3B and 8B LLMs (Table 1,2), which somehow fits the scope of the mentioned "large reasoning models".
>
> We initially focus on the math domain, the most widely used testbed for PRMs [8,9,10,11], where CRM consistently demonstrates its effectiveness in Best-of-N sampling, beam search, and RL training.
>
> Your suggestion offers an additional perspective for validating CRM. Accordingly, we expanded it to domains like health and physics.
> Following the setup in [12], we evaluate CRM, PRM and PQM on BoN sampling experiment across various domains.
> As shown in Table 1 below, CRM outperforms the baselines in almost all tested domains, underscoring the effectiveness of our approach across diverse domains.
> We have added the full experimental results and implementation details to the revised manuscript (Section 4.5.4).
>
> Table 1: Best-of-N accuracy of different methods across multiple domains.
> |BoN Accuracy | Biology | | | Business | | | Health | | | History | | | Physics | | |
> | :--- | :---: | :---: | :---: | :---: | :---: | :---: | :---: | :---: | :---: | :---: | :---: | :---: | :---: | :---: | :---: |
> | **Methods** | @16 | @32 | @64 | @16 | @32 | @64 | @16 | @32 | @64 | @16 | @32 | @64 | @16 | @32 | @64 |
> | PRM | 80.0 | 79.2 | 79.2 | 64.8 | 64.8 | 66.2 | 60.7 | 57.1 | 57.1 | 48.0 | 48.7 | 48.7 | 61.0 | 58.9 | 60.3 |
> | PQM | 80.8 | 80.0 | 80.8 | 64.1 | 63.5 | 65.5 | 62.1 | 61.4 | 61.4 | 50.0 | 51.3 | 49.3 | 61.6 | 58.2 | 62.3 |
> | **CRM (ours)** | 83.1 | 83.1 | 82.3 | 66.2 | 66.9 | 68.2 | 61.4 | 62.1 | 60.7 | 50.7 | 52.0 | 51.3 | 64.4 | 63.7 | 66.4 |
>
> [1] Daya Guo, Dejian Yang, Haowei Zhang, Junxiao Song, Ruoyu Zhang, Runxin Xu, Qihao Zhu, Shirong Ma, Peiyi Wang, Xiao Bi, et al. Deepseek-r1: Incentivizing reasoning capability in llms via reinforcement learning. arXiv preprint arXiv:2501.12948, 2025.
>
> [2] Weihao Zeng, Yuzhen Huang, Qian Liu, Wei Liu, Keqing He, Zejun MA, and Junxian He. SimpleRL-zoo: Investigating and taming zero reinforcement learning for open base models in the wild. In Second Conference on Language Modeling, 2025.
>
> [3] Jie Cheng, Gang Xiong, Ruixi Qiao, Lijun Li, Chao Guo, Junle Wang, Yisheng Lv, and Fei-Yue Wang. Stop summation: Min-form credit assignment is all process reward model needs for reasoning. In The Thirty-ninth Annual Conference on Neural Information Processing Systems, 2025.
>
> [4] Yuhao Wang, Xiaopeng Li, Cheng Gong, Ziru Liu, Suiyun Zhang, Rui Liu, and Xiangyu Zhao. Efficient reasoning via reward model. arXiv preprint arXiv:2511.09158, 2025.
>
> [5] Leitian Tao, Ilia Kulikov, Swarnadeep Saha, Tianlu Wang, Jing Xu, Yixuan Li, Jason E Weston, and Ping Yu. Hybrid reinforcement: When reward is sparse, it's better to be dense. arXiv preprint arXiv:2510.07242, 2025.
>
> [6] Fengli Xu, Qianyue Hao, Zefang Zong, Jingwei Wang, Yunke Zhang, Jingyi Wang, Xiaochong Lan, Jiahui Gong, Tianjian Ouyang, Fanjin Meng, et al. Towards large reasoning models: A survey of reinforced reasoning with large language models. arXiv preprint arXiv:2501.09686, 2025.
>
> [7] Kaiyan Zhang, Yuxin Zuo, Bingxiang He, Youbang Sun, Runze Liu, Che Jiang, Yuchen Fan, Kai Tian, Guoli Jia, Pengfei Li, et al. A survey of reinforcement learning for large reasoning models. arXiv preprint arXiv:2509.08827, 2025.
>
> [8] Jie Cheng, Gang Xiong, Ruixi Qiao, Lijun Li, Chao Guo, Junle Wang, Yisheng Lv, and Fei-Yue Wang. Stop summation: Min-form credit assignment is all process reward model needs for reasoning. In The Thirty-ninth Annual Conference on Neural Information Processing Systems, 2025.
>
> [9] Wendi Li and Yixuan Li. Process reward model with q-value rankings. In The Thirteenth International Conference on Learning Representations, 2025.
>
> [10] Lifan Yuan, Wendi Li, Huayu Chen, Ganqu Cui, Ning Ding, Kaiyan Zhang, Bowen Zhou, Zhiyuan Liu, and Hao Peng. Free process rewards without process labels. In Forty-second International Conference on Machine Learning, 2025.
>
> [11] Peiyi Wang, Lei Li, Zhihong Shao, Runxin Xu, Damai Dai, Yifei Li, Deli Chen, Yu Wu, and Zhi-fang Sui. Math-shepherd: Verify and reinforce LLMs step-by-step without human annotations. InLun-Wei Ku, Andre Martins, and Vivek Srikumar (eds.), Proceedings of the 62nd Annual Meet-ing of the Association for Computational Linguistics.
>
> [12] Thomas Zeng, Shuibai Zhang, Shutong Wu, Christian Classen, Daewon Chae, Ethan Ewer, Minjae Lee, Heeju Kim, Wonjun Kang, Jackson Kunde, Ying Fan, Jungtaek Kim, Hyung Il Koo, Kannan Ramchandran, Dimitris Papailiopoulos, and Kangwook Lee. VersaPRM: Multi-domain process reward model via synthetic reasoning data. In Forty-second International Conference on Machine Learning, 2025.

---

> > ### Comment · Reviewer_8bhE · 2025-11-27
> >
> > The authors have addressed most of my concerns in their rebuttal; therefore, I will revise my scores accordingly.

---

> > > ### Author Response · Authors · 2025-11-28
> > > **Response to Reviewer 8bhE**
> > >
> > > Thank you for raising your score.
> > > We made a deliberate effort to address each point comprehensively and to provide clear evidence and explanations.
> > > We are glad to hear that your concerns have been thoroughly addressed, and we appreciate your positive reassessment.

---

### Official Review · Reviewer_tScJ · 2025-10-31

**Soundness:** 4
**Presentation:** 4
**Contribution:** 3
**Rating:** 8
**Confidence:** 4

**Summary:**

This paper proposes Conditional Reward Modeling (CRM) that frames LLM reasoning as a temporal process leading to a correct answer. The core idea is to model the reasoning process as a conditional probability process, associating the reward of each step with the preceding tokens and the final result. The author conducted experiments on various applications of their reward model, in most of which they showed good performance. The cost of this article is also acceptable, and the experimental comparison is fair from this perspective. The current improvement in the reward model mostly comes from the reasoning reward model, which is easily hacked, or training data scaling. I think this is a good paper that should be accepted as a poster.

**Strengths:**

1. The paper is well written, and the proposed method is reasonable.
2. The experiment is sufficient to demonstrate the effectiveness of the method.
3. The experiment can prove that the reward learned by the method is robust and is not easily hacked.

**Weaknesses:**

The author can try to expand the method to more fields, such as code, or even fields without verifiers.

**Questions:**

1. The results in Figure 4 show the behavior of reward hacking. But the reviewer has conducted a similar experiment using Math-Shepherd, and although the effect was not good, the AIME2024 accuracy did not decrease to 0. Is there a problem with this part of the experiment? That is, the author obtained a good training hyperparameter on CRM and directly used it on other baselines, which require different hyperparameters.
2. In the BoN exps, the authors use the score S(T) for CRM and take the min value for other models. What would happen if other models also use S(T)?

---

> ### Author Response · Authors · 2025-11-22
> **Author Response to Reviewer tScJ (1/2)**
>
> We appreciate the reviewer's careful reading and insightful comments.
>
> > Weaknesses: The author can try to expand the method to more fields, such as code, or even fields without verifiers.
>
> Thank you for your helpful suggestion.
>
>
> We initially focused on the math domain, the most established testbed for PRMs in prior works [1,2,3,4].
> In this setting, CRM consistently improves Best-of-N (BoN) sampling, beam search, and RL training performance, providing strong evidence for the effectiveness of our CRM.
>
> Your suggestion provides an additional perspective for validating our method, so we expanded our evaluation to more domains such as health and physics domains.
>
> Following the experimental and data settings in [5], we evaluated CRM and the two strong baselines (PRM and PQM) used in our paper on BoN sampling experiment across various domains  using Qwen2.5-3B-Instruct as the backbone.
> As shown in Table 1 below, our CRM outperforms the baselines in almost all tested domains, demonstrating that our modeling approach remains effective across diverse domains.
> We have added the full experimental results and implementation details to the revised manuscript (Section 4.5.4), where all new changes are marked in blue.
>
> Table 1: Best-of-N accuracy of different methods across multiple domains.
> |BoN Accuracy | Biology | | | Business | | | Health | | | History | | | Physics | | |
> | :--- | :---: | :---: | :---: | :---: | :---: | :---: | :---: | :---: | :---: | :---: | :---: | :---: | :---: | :---: | :---: |
> | **Methods** | @16 | @32 | @64 | @16 | @32 | @64 | @16 | @32 | @64 | @16 | @32 | @64 | @16 | @32 | @64 |
> | PRM | 80.0 | 79.2 | 79.2 | 64.8 | 64.8 | 66.2 | 60.7 | 57.1 | 57.1 | 48.0 | 48.7 | 48.7 | 61.0 | 58.9 | 60.3 |
> | PQM | 80.8 | 80.0 | 80.8 | 64.1 | 63.5 | 65.5 | 62.1 | 61.4 | 61.4 | 50.0 | 51.3 | 49.3 | 61.6 | 58.2 | 62.3 |
> | **CRM (ours)** | 83.1 | 83.1 | 82.3 | 66.2 | 66.9 | 68.2 | 61.4 | 62.1 | 60.7 | 50.7 | 52.0 | 51.3 | 64.4 | 63.7 | 66.4 |
>
>
>
> > Q1: The results in Figure 4 show the behavior of reward hacking. But the reviewer has conducted a similar experiment using Math-Shepherd, and although the effect was not good, the AIME2024 accuracy did not decrease to 0. Is there a problem with this part of the experiment? That is, the author obtained a good training hyperparameter on CRM and directly used it on other baselines, which require different hyperparameters.
>
> Thank you for your efforts.
> We first clarify that hyperparameter tuning was performed for other baselines; however, similar reward hacking behavior persisted across different settings.
>
> Below, we explain the discrepancy.
> Our experimental results are not in conflict with your observation. In fact, the observed gap is reasonable and may arise from the following two factors:
>
> - (i) **Min-max scaling**: In Figure 4(a), min-max scaling causes the large rewards at later steps to overshadow the early spike, obscuring the onset of reward hacking.
> In reality, both PRM and PQM exhibit reward hacking around step 50, during which AIME2024 accuracy fluctuates and gradually declines.
> This pattern is similar to the reviewer's observation.
> - (ii) **Extended training**: Ideally, training should stop after reward hacking emerges, but we continued training for a fair comparison with CRM. This led to more severe reward hacking, including repetitive generations (see Figure 4(c) and Appendix C in our paper), making responses unusable and answers hard to extract, which ultimately caused accuracy to drop to zero.
>
> Another possibility is that the model behaviors under reward hacking encountered by the reviewer differ from ours.
> In exploratory experiments, we observed that model behaviors resulting from reward hacking do not always manifest identically, as they are influenced by multiple factors.
> In some cases, the model produces specific response patterns that lead to non-zero accuracy.
> Understanding how reward hacking alters model behavior is a promising direction.
> We can design experiments to explore how factors like hyperparameters, training data diversity and quality, and task complexity influence different reward hacking behaviors.
> Internally, we can investigate differences in the model's neuron during various reward hacking behaviors.
> We consider these as promising future work.
> We again thank the reviewer for insightful experiments.

---

> ### Author Response · Authors · 2025-11-22
> **Author Response to Reviewer tScJ (2/2)**
>
> > Q2. In the BoN exps, the authors use the score S(T) for CRM and take the min value for other models. What would happen if other models also use S(T)?
>
> Thank you for your valuable feedback.
>
> We adopt the minimum value to aggregate step rewards in BoN sampling for baselines following their original implementations [2,3]. In this setting, a low-reward (possibly indicating incorrect) step dominates the final score of the whole response, since an incorrect step can invalidate the entire solution in math reasoning.
>
> Recall that our CRM defines $S(T)$ as the probability of maintaining correct reasoning up to step $T$, which serves as an innovative step reward aggregation scheme using the probabilistic chain rule, as formulated in Eq. (5) in our paper.
>
> To give baseline methods a similar $S(T)$-style aggregation, we applied product, sum, and average over their step rewards, and evaluated these variants via BoN sampling on MATH500 using Qwen2.5-3B-Instruct as the backbone.
>
>
> As shown in Table 2 below, these variants still underperform our CRM using $S(T)$ and even their original min-aggregation versions.
> This is consistent with the analysis and results in [1,5] that min-aggregation performs better.
> The observation indicates that simply aggregating the process rewards, i.e., the step rewards estimated by other process reward models, does not work well, reflecting the effectiveness of our CRM method.
>
>
>
>
> Table 2: Best-of-N accuracy on MATH500 using different aggregation methods.
>
> | BoN Accuracy|  | MATH500 | | | | |
> | :--- | :--- | :--- | :--- | :--- | :--- | :--- |
> |**Methods** |**Aggregation** | @8 | @16 | @32 | @64 | @128 |
> | PRM | min     | 54.2 | 55.2 | 55.2 | 54.4 | 54.6 |
> | PRM | sum     | 46.4 | 46.2 | 43.2 | 43.8 | 43.6 |
> | PRM | mean    | 48.6 | 50.6 | 51.8 | 53.4 | 52.2 |
> | PRM | product | 53.0 | 54.6 | 54.8 | 54.6 | 53.2 |
> | PQM | min     | 53.2 | 54.4 | 54.8 | 54.8 | 55.8 |
> | PQM | sum     | 49.4 | 49.8 | 47.6 | 46.4 | 43.4 |
> | PQM | mean    | 52.2 | 52.4 | 52.4 | 51.4 | 51.8 |
> | PQM | product | 50.6 | 54.2 | 55.2 | 52.8 | 53.0 |
> | CRM (ours)| S(T)|53.0| 56.4 | 56.6 | 55.8 | 56.6 |
>
>
> [1] Jie Cheng, Gang Xiong, Ruixi Qiao, Lijun Li, Chao Guo, Junle Wang, Yisheng Lv, and Fei-Yue Wang. Stop summation: Min-form credit assignment is all process reward model needs for reasoning. In The Thirty-ninth Annual Conference on Neural Information Processing Systems, 2025.
>
> [2] Wendi Li and Yixuan Li. Process reward model with q-value rankings. In The Thirteenth International Conference on Learning Representations, 2025.
>
> [3] Lifan Yuan, Wendi Li, Huayu Chen, Ganqu Cui, Ning Ding, Kaiyan Zhang, Bowen Zhou, Zhiyuan Liu, and Hao Peng. Free process rewards without process labels. In Forty-second International Conference on Machine Learning, 2025.
>
> [4] Peiyi Wang, Lei Li, Zhihong Shao, Runxin Xu, Damai Dai, Yifei Li, Deli Chen, Yu Wu, and Zhi-fang Sui. Math-shepherd: Verify and reinforce LLMs step-by-step without human annotations. In Lun-Wei Ku, Andre Martins, and Vivek Srikumar (eds.), Proceedings of the 62nd Annual Meet-ing of the Association for Computational Linguistics.
>
> [5] Thomas Zeng, Shuibai Zhang, Shutong Wu, Christian Classen, Daewon Chae, Ethan Ewer, Minjae Lee, Heeju Kim, Wonjun Kang, Jackson Kunde, Ying Fan, Jungtaek Kim, Hyung Il Koo, Kannan Ramchandran, Dimitris Papailiopoulos, and Kangwook Lee. VersaPRM: Multi-domain process reward model via synthetic reasoning data. In Forty-second International Conference on Machine Learning, 2025.

---

> > ### Comment · Reviewer_tScJ · 2025-11-24
> > **Reviewer Reasponse**
> >
> > Thank you for your response. I think the paper is suitable for a poster acceptance, and I will maintain my current score accordingly.

---

> > > ### Author Response · Authors · 2025-11-26
> > > **Response to Reviewer tScJ**
> > >
> > > Thank you very much for your recognition of our paper. We are glad that our responses helped clarify your questions, and we sincerely appreciate your thorough review and constructive feedback.

---

### Official Review · Reviewer_G7br · 2025-10-31

**Soundness:** 2
**Presentation:** 3
**Contribution:** 3
**Rating:** 6
**Confidence:** 3

**Summary:**

The paper introduces Conditional Reward Modeling (CRM) for multi-step reasoning, aligning step-level rewards with the final outcome through a principled probabilistic framing centered on the “first error” in a chain of thought. This yields a single, comparable trajectory score that supports better sample selection (Best-of-N), stronger beam-search guidance, and more stable RL training with reduced reward hacking, while remaining complementary to verifiable outcome rewards when available. Experiments on math reasoning benchmarks and multiple backbones show consistent gains over PRM/PQM/IPRM/ORM baselines and clearer localization of the first wrong step. Ablations indicate robustness to limited step-level supervision and to reward-shaping choices, suggesting a simple and practical framework for outcome-aligned process rewards.

**Strengths:**

The paper poses an interesting problem and ties the method well to practical RL applications.

**Weaknesses:**

The experiments do not appear to discuss the latest PRM (including trainable PRM) designs. [1][2]

CRM seems to converge with small n; please add Major@N and Pass@N baselines to Table 1 to contextualize performance and the gap to an upper bound.

Table 3 should include an RLVR baseline.

[1] R-PRM: Reasoning-Driven Process Reward Modeling

[2] GenPRM: Scaling Test-Time Compute of Process Reward Models via Generative Reasoning

**Questions:**

N/A

---

> ### Author Response · Authors · 2025-11-22
> **Author Response to Reviewer G7br (1/3)**
>
> We thank the reviewer for the valuable time and constructive feedback.
>
> > Weakness 1: The experiments do not appear to discuss the latest PRM (including trainable PRM) designs. [1][2]
>
> Thank you for your suggestion.
>
> Both work [1,2] aims at incorporating reasoning capabilities into reward modeling by conducting a thinking process before reward scoring, which provide a more explainable reward estimation though, sharing similar problems with the existing works discussed in our paper (e.g., lacking a connection from step to final outcome rewards).
>
> These works are *orthogonal* to our proposed method.
> Recall that our CRM model directly assigns rewards through a novel conditional probability estimation and derive the final outcome reward via the probability chain rule, which can further integrate the thinking process proposed in [1,2] to enhance the reward modeling and estimation.
> We believe this is worth further exploration as a promising future work.
> Thank you for your valuable comments!
>
> Regarding your suggestions, while also considering the time limit of rebuttal constraints, we conducted an experimental comparison with [1], following the settings of Best-of-N sampling on MATH500 described in Section 4.2 of our paper.
> The results are detailed in Table 1 below, demonstrating that our CRM surpasses the latest R-PRM [1] model in terms of the sample rewarding.
> We will conduct the full experiment in the future paper revision.
>
> For this revision, we have (i) revised the discussion of related work in Section 2 to reflect the above points (Line 121-124); we have also (ii) added these additional experiments in Appendix H of the revised manuscript.
> We have marked all changes in the revised manuscript in blue.
>
> Table 1: Comparison to R-PRM on Best-of-N sampling experiment on MATH500.
> | BoN Accuracy | Best-of-8 | Best-of-16 |
> |---------|-----------|------------|
> | R-PRM [1]      | 49.8      | 51.2       |
> | **CRM (ours)**  | 53.0      | 56.4       |
>
> [1] Shuaijie She, Junxiao Liu, Yifeng Liu, Jiajun Chen, Xin Huang, and Shujian Huang. 2025. R-PRM: Reasoning-Driven Process Reward Modeling. In Proceedings of the 2025 Conference on Empirical Methods in Natural Language Processing, pages 13449–13462, Suzhou, China. Association for Computational Linguistics.
>
> [2] Jian Zhao, Runze Liu, Kaiyan Zhang, Zhimu Zhou, Junqi Gao, Dong Li, Jiafei Lyu, Zhouyi Qian, Biqing Qi, Xiu Li, et al. Genprm: Scaling test-time compute of process reward models via generative reasoning. arXiv preprint arXiv:2504.00891, 2025.

---

> ### Author Response · Authors · 2025-11-22
> **Author Response to Reviewer G7br (2/3)**
>
> > Weakness 2.1: CRM seems to converge with small n
>
> Thank you for the insightful observation.
> We would like to clarify that the convergence pattern is largely determined by the LLM response generator used in the Best-of-N (BoN) sampling experiment.
>
> Recall that, in BoN, the LLM first samples $N$ candidate responses for each query.
> The reward model then selects the response with the highest predicted reward.
> Therefore, the behavior of BoN directly reflects the sampling characteristics of the underlying LLM response generator, as discussed below.
>
> - (i) *Weaker LLM generators* have a low probability of producing correct responses. At small $N$, they may generate few correct candidates, leaving the reward model with nothing correct to select. As $N$ increases, the sampled candidates are more likely to include correct answers, so the BoN performance improves and eventually stabilizes.
> - (ii) *Stronger LLM generators* (in which case our experiment setting more probably falls) are able to produce correct responses even with small $N$. In this case, the BoN process is not limited by candidate quality, leading to faster convergence. This also explains why other baselines in Table 1 of the paper show similar behavior and converge at small $N$.
>
> Overall, the convergence of CRM at small $N$ is a direct consequence of the LLM generator's sampling quality, rather than a limitation of CRM itself.
>
> This analysis could be further supported by the **Pass@N** metric you mentioned, which measures the probability that the LLM generator produces *at least one* correct response among the $N$ sampled responses [1].
> Pass@N therefore serves as an upper bound on the achievable performance of BoN sampling with any reward model.
> In Table 2 shown below, we report the Pass@N of two LLM generators: a relatively weaker LLM generator MetaMath-Mistral-7B, and a stronger LLM generator LLaMA-3.1-8B-Instruct, which is the generator used in our main experiments.
> Two observations emerge:
> - (i) The weaker generator MetaMath-Mistral-7B shows a substantial gap between Pass@8 and Pass@128. Its achievable upper bound under the BoN setting continues to grow with larger $N$, with *no early convergence* at small $N$.
> - (ii) The stronger LLM generator LLaMA-3.1-8B-Instruct, in contrast, achieves high Pass@N even for small $N$, leading to early convergence.
>
> Together, these findings reinforce our earlier analysis: the convergence behavior of CRM under BoN is largely determined by the sampling quality of the underlying LLM generator.
>
> Table 2: Pass@N of different LLM generators on MATH500 and GSM-Plus datasets.
> | Pass@N |  MATH500 | | | | | | GSM-Plus | | | | | |
> | :--- |  :--- | :--- | :--- | :--- | :--- | :--- | :--- | :--- | :--- | :--- | :--- | :--- |
> | LLM Generator | Δ[Pass@8→@128] | Pass@8 | Pass@16 | Pass@32 | Pass@64 | Pass@128 | Δ[Pass@8→@128] | Pass@8 | Pass@16 | Pass@32 | Pass@64 | Pass@128 |
> | MetaMath-Mistral-7B  | **27.4** | 53.0 | 62.0 | 69.6 | 75.8 | 80.4 | **11.1** | 73.2 | 78.1 | 80.5 | 83.0 | 84.3 |
> | Llama-3.1-8B-Instruct  | **16.4** | 73.8 | 79.2 | 84.6 | 87.4 | 90.2 | **6.3** | 79.2 | 81.2 | 82.2 | 83.3 | 85.5 |
>
>
> Motivated by your comment, we further evaluated CRM using the weaker MetaMath-Mistral-7B generator and present the BoN performance in Table 3.
> As expected, CRM continues to improve as $N$ increases, aligning well with the Pass@N trends and supporting our explanation above.
>
> We have added the above experiments and analysis to Appendix I of the revised manuscript. Thank you again for your insightful comments.
>
>
> Table 3. BoN performance of our CRM on MATH500 using MetaMath-Mistral-7B as the LLM generator.
> | BoN Accuracy |  | MATH500 | | | | |
> | :--- | :--- | :--- | :--- | :--- | :--- | :--- |
> |**Method** | **Generator** | 8 | 16 | 32 | 64 | 128 |
> | CRM | MetaMath-Mistral-7B | 35.6 | 39.0 | 39.2 | 40.2 | 41.6 |
>
>
> > Weakness 2.2: please add Major@N and Pass@N baselines to Table 1 to contextualize performance and the gap to an upper bound.
>
> We thank the reviewer for suggesting these metrics.
>
> We have added the Major@N and Pass@N baselines to Table 1 in the revised manuscript. These metrics more faithfully reflect the model capability and demonstrate that our CRM significantly outperforms the majority voting baseline, which selects the most frequent answer among $N$ responses, and closely approaches the oracle performance.

---

> ### Author Response · Authors · 2025-11-22
> **Author Response to Reviewer G7br (3/3)**
>
> > Weakness 3: Table 3 should include an RLVR baseline.
>
> Thank you for the helpful suggestion.
>
> We have added the RLVR baseline (abbreviated as VR for consistency with our paper) in the revised manuscript (Line 376), and the corresponding results are reported in Table 4 below.
> For VR, the policy is trained using only final-answer verifiable rewards, while keeping all other settings identical to our main experiments.
>
>
> Notably, CRM consistently outperforms the VR baseline, and additional gains can be achieved by combining CRM with VR.
> These findings demonstrate that CRM is effective on its own, without relying on ground-truth verification.
> Moreover, when VR is available, CRM offers complementary rather than redundant supervision.
>
>
>
> Table 4: Pass@1 accuracy evaluated on mathematical reasoning benchmarks.
> | | MATH500 | Minerva Math | OlympiadBench | AIME25 | AIME24 | AMC23 |
> | :--- | :--- | :--- | :--- | :--- | :--- | :--- |
> | VR | 76.2 | 38.6 | 38.0 | 16.6 | 30.0 | 62.5 |
> | CRM | 77.8 | 40.0 | 39.3 | 23.3 | 43.3 | 67.5 |
> | CRM+VR | 80.4 | 43.0 | 42.1 | 26.6 | 33.3 | 72.5 |

---

> > ### Comment · Reviewer_G7br · 2025-11-27
> >
> > Thank you very much for the authors' response. I still have some concerns:
> >
> > * Majority@N: Why does the performance decrease as N increases? There seems to be an issue here.
> > * As can be seen in Table 1, the Pass@N performance upper bound remains high and even increases with N. This indicates that more correct answers are being sampled and selected. The authors still cannot explain why CRM directly converges and fails to improve further.

---

> > > ### Author Response · Authors · 2025-12-03
> > > **Response to Reviewer G7br's Follow-Up Questions (1/3)**
> > >
> > > We appreciate your careful reading and your constructive follow-up questions.
> > >
> > > > Majority@N: Why does the performance decrease as N increases? There seems to be an issue here.
> > >
> > > We would like to clarify that such a decrease is normal and reasonable.
> > >
> > > **For hard queries (where the LLM is more likely to generate incorrect answers), major@N decreases as N increases [1].**
> > > The assumption behind major@N increasing with larger $N$ is that "model favors the correct answer and generates it more frequently than incorrect ones" [2].
> > > However, this assumption does not always hold [1, 3, 4, 5, 6].
> > > Whether the assumption holds depends on the query difficulty [1], which is determined by both the query itself and the LLM generator used to answer it.
> > > As noted in [1], increasing $N$ can improve major@N on easy queries but may degrade it on hard queries.
> > >
> > >
> > > **Our task is hard (containing many hard queries), and therefore major@N decreases as N increases.**
> > > We follow the procedure proposed in [1] to define easy and hard queries in our experimental setting.
> > > Specifically, we generate 128 responses for each query using the same LLM generator as in our original experiments and compute its empirical correctness rate.
> > > We classify a query as *easy* if its correctness rate exceeds 0.5, and as *hard* if it falls below 0.5.
> > > We then compute both the size proportion and the average correctness rate of the easy and hard query sets.
> > > These results are used for the computation in [1] and are shown in Table 1 below.
> > > According to the calculation and conclusion of Theorem 2 in [1], our task is hard, where major@N decreases as $N$ increases.
> > >
> > >
> > > Table 1: Size proportions and average correctness rate of easy and hard queries in the MATH500 dataset.
> > > | MATH500     | Easy queries   | Hard queries   |
> > > |-------------|--------|--------|
> > > | size proportion  | 44.8%  | 55.2%  |
> > > | average correctness rate |0.808 | 0.156|
> > >
> > >
> > > **Results from prior work and publicly available datasets also exhibit a similar decrease.**
> > > - *Evidence from prior work*. As shown in [1, 5], major@N performance often saturates or even degrades as $N$ increases.
> > > In addition, [7] reports that on GSM8K, MATH-500, and Gaokao2023, most major@N scores decrease or remain nearly unchanged as $N$ grows.
> > > - *Evidence from public datasets*. We compute major@N on the three public test sets from [8], each generated by a different LLM.
> > > As shown in Table 2 below, these major@N results also display a decrease.
> > >
> > > These results collectively indicate that the decrease in major@N is normal and reasonable.
> > >
> > > Table 2: The major@N statistics on public datasets.
> > > | Major@N | GSM-Plus |  | | | | MATH500| | | | |
> > > | :--- | :--- | :--- | :--- | :--- | :--- | :--- | :--- | :--- | :--- | :--- |
> > > | LLM Generator | @8 | @16 | @32 | @64 | @128 | @8 | @16 | @32 | @64 | @128 |
> > > | MetaMathMistral-7B | 51.3 | 50.3 | 49.8 | 49.4 | 49.4 | 24.0 | 21.4 | 20.6 | 20.0 | 20.2 |
> > > | MuggleMath-13B | 46.7 | 45.6 | 43.9 | 43.3 | 43.2 | 15.6 | 15.0 | 14.8 | 14.0 | 13.0 |
> > > | Llama-3-70B-Instruct | 73.8 | 73.0 | 72.5 | 72.3 | 72.7 | 41.0 | 40.0 | 38.4 | 37.4 | 37.2|

---

> > > ### Author Response · Authors · 2025-12-03
> > > **Response to Reviewer G7br's Follow-Up Questions (3/3)**
> > >
> > > [1] Lingjiao Chen, Jared Davis, Boris Hanin, Peter Bailis, Ion Stoica, Matei Zaharia, and James Zou. Are more llm calls all you need? towards the scaling properties of compound ai systems. In Advances in Neural Information Processing Systems, volume 37, pp. 45767–45790, 2024.
> > >
> > > [2] Amin Rakhsha, Kanika Madan, Tianyu Zhang, Amir-massoud Farahmand, and Amir Khasahmadi. Majority of the bests: Improving best-of-n via bootstrapping. arXiv preprint arXiv:2511.18630, 2025.
> > >
> > > [3] Zhangyue Yin, Qiushi Sun, Qipeng Guo, Zhiyuan Zeng, Xiaonan Li, Tianxiang Sun, Cheng Chang, Qinyuan Cheng, Ding Wang, Xiaofeng Mou, et al. Aggregation of reasoning: A hierarchical framework for enhancing answer selection in large language models. In Proceedings of the 2024 Joint International Conference on Computational Linguistics, Language Resources and Evaluation (LREC-COLING 2024), pp. 609–625, 2024.
> > >
> > > [4] Danqing Wang, Jianxin Ma, Fei Fang, and Lei Li. Typedthinker: Diversify large language model reasoning with typed thinking. arXiv preprint arXiv:2410.01952, 2024.
> > >
> > > [5] Yichao Fu, Xuewei Wang, Yuandong Tian, and Jiawei Zhao. Deep think with confidence. arXiv preprint arXiv:2508.15260, 2025
> > >
> > > [6] Wenting Zhao, Pranjal Aggarwal, Swarnadeep Saha, Asli Celikyilmaz, Jason Weston, and Ilia Kulikov. The majority is not always right: Rl training for solution aggregation. arXiv preprint arXiv:2509.06870, 2025.
> > >
> > > [7] Wenxiang Chen, Wei He, Zhiheng Xi, Honglin Guo, Boyang Hong, Jiazheng Zhang, Nijun Li, Tao Gui, Yun Li, Qi Zhang, et al. Better process supervision with bi-directional rewarding signals. In Findings of the Association for Computational Linguistics: ACL 2025, pp. 14471–14485, 2025.
> > >
> > > [8] Wendi Li and Yixuan Li. Process reward model with q-value rankings. In The Thirteenth International Conference on Learning Representations, 2025.
> > >
> > > [9] Audrey Huang, Adam Block, Qinghua Liu, Nan Jiang, Akshay Krishnamurthy, and Dylan J Foster. Is best-of-n the best of them? coverage, scaling, and optimality in inference-time alignment. arXiv preprint arXiv:2503.21878, 2025.
> > >
> > > [10] Benedikt Stroebl, Sayash Kapoor, and Arvind Narayanan. Inference scaling flaws: The limits of llm resampling with imperfect verifiers. arXiv preprint arXiv:2411.17501, 2024.
> > >
> > > [11] Hyung Gyu Rho and Sian Lee. Best of mini-n in-loop sampling: A contextual quality reward model for reliable and efficient best-of-n sampling. arXiv preprint arXiv:2510.04087, 2025.

---

> ### Author Response · Authors · 2025-12-03
> **Response to Reviewer G7br's Follow-Up Questions (2/3)**
>
> > As can be seen in Table 1, the Pass@N performance upper bound remains high and even increases with N. This indicates that more correct answers are being sampled and selected. The authors still cannot explain why CRM directly converges and fails to improve further.
>
> We thank the reviewer for the comment, which prompted us to further investigate Best-of-N sampling mechanisms.
>
> First, we must emphasize that pass@N represents an upper bound rather than a sampling method, and therefore an increase in pass@N does *not* indicate that "more correct answers are being selected", as stated by the reviewer.
> The phenomenon noted by the reviewer is due to the inherent gap between pass@N and Best-of-N (BoN) performance, as well as the properties of the specific LLM generator.
>
> **Whether further improvement occurs depends on the LLM generator used. Our CRM shows an increasing trend in BoN performance as $N$ increases when using a different LLM generator.**
> As discussed in our earlier response, when we switch to a weaker generator (MetaMath-Mistral-7B), our CRM shows an increase in BoN performance as $N$ grows. We further extended our experiments, and the results are summarized in Table 3 below.
> The results align with our previous conclusion:
> with a strong generator (Llama-3.1-8B-Instruct), all methods converge at small $N$ and show no further improvement, whereas with a weak generator (MetaMath-Mistral-7B), all methods exhibit increasing BoN performance as $N$ increases.
> This difference stems from the choice of LLM generator in the experimental setup.
> We have added the experiments under the different experimental setup to Appendix I in the revised manuscript.
>
> **The underlying reason is that different LLM generators induce different optimal $N$ [10], beyond which BoN performance no longer improves [9].**
> The LLM generator used in our earlier experiments has a small optimal $N$, causing all methods to converge early and fail to improve further.
> The existence of such an optimal $N$ arises from a **trade-off** [9]: larger $N$ increases the probability of sampling higher-quality correct responses, but it also increases the likelihood of obtaining low-quality incorrect responses with spuriously high rewards [9].
> Moreover, the optimal $N$ varies across different LLM generators [10].
> This aligns with the experimental results in Table 3 below: with a strong generator (Llama-3.1-8B-Instruct), high-quality responses are easier to obtain, leading to a smaller optimal $N$, and therefore all methods converge at small $N$.
> In contrast, with a weak generator (MetaMath-Mistral-7B), the optimal $N$ is larger, so BoN performance continues to improve for $N$ < 128.
>
>
> **The fundamental gap between pass@N and BoN implies that BoN performance does not necessarily follow pass@N.**
> As the LLM generator is allowed more attempts (larger $N$), pass@N naturally increases.
> While pass@N reflects the oracle upper bound, BoN performance is harder to reach this oracle for two main reasons:
> (i) BoN requires scoring all responses to select the best one, while pass@N only checks for one correct response, making BoN much harder. Noisy outputs from the LLM generator further complicate BoN.
> (ii) A domain discrepancy exists between the reward model's training data and the responses generated by the LLM generator, thereby introducing generalization challenges.
> This indicates that all current reward models still have room for improvement.
> Prior studies [7, 9, 10, 11] also report that BoN performance often converges as $N$ increases, and [2, 9, 10, 11] analyze the limitations of BoN sampling and propose improvements.
> We believe these directions are orthogonal to our modeling approach, and can be seamlessly integrated with our CRM, but they are beyond the scope of this work.
> In future work, we plan to explore training strategies tailored to BoN's characteristics to narrow the gap between BoN and the oracle.
>
> We thank the reviewer again for the comments.
>
>
>
>
> Table 3. Best-of-N sampling performance across methods using different LLM generators.
> | LLM Generator            | Method | MATH500   |   |   |   |      |
> |--------------------------|--------|------|------|------|------|-------|
> |                          |        | @8   | @16  | @32  | @64  | @128 |
> | MetaMath-Mistral-7B      | Pass@N | 53.0 | 62.0 | 69.6 | 75.8 | 80.4  |
> |                          | PRM    | 34.0 | 35.8 | 35.2 | 37.4 | 38.0  |
> |                          | PQM    | 36.4 | 38.4 | 37.8 | 38.6 | 40.4  |
> |                          | CRM (ours) | 35.6 | 39.0 | 39.2 | 40.2 | 41.6  |
> | Llama-3.1-8B-Instruct    | Pass@N | 73.8 | 79.2 | 84.6 | 87.4 | 90.2  |
> |                          | PRM    | 54.2 | 55.2 | 55.2 | 54.2 | 54.6  |
> |                          | PQM    | 53.2 | 54.4 | 54.8 | 54.8 | 55.8  |
> |                          | CRM (ours) | 53.0 | 56.4 | 56.6 | 55.8 | 56.6  |

---

> ### Author Response · Authors · 2025-12-03
> **Supplementary Response to Reviewer G7br (Weakness 1)**
>
> Regarding the two baselines [1][2] mentioned in Weakness 1, our initial response included only results for R-PRM [1] due to time constraints.
> **We are now pleased to report that we have completed the reproduction of GenPRM [2]** and have updated Table 1 accordingly.
>
> The updated results are consistent with the conclusion in our previous response, demonstrating that our CRM surpasses the latest PRM in terms of the sample rewarding.
>
> We have added these additional experiments in Appendix H of the revised manuscript.
> We again thank the reviewer for the helpful suggestion.
>
> Table 1: Comparison with R-PRM and GenPRM on Best-of-N sampling experiments on MATH500.
> | BoN Accuracy | Best-of-8 | Best-of-16 |
> |---------|-----------|------------|
> | R-PRM [1]       | 49.8      | 51.2       |
> | GenPRM [2]      | 50.8      | 55.2       |
> | **CRM (ours)**  | 53.0      | 56.4       |
>
> [1] Shuaijie She, Junxiao Liu, Yifeng Liu, Jiajun Chen, Xin Huang, and Shujian Huang. 2025. R-PRM: Reasoning-Driven Process Reward Modeling. In Proceedings of the 2025 Conference on Empirical Methods in Natural Language Processing, pages 13449–13462, Suzhou, China. Association for Computational Linguistics.
>
> [2] Jian Zhao, Runze Liu, Kaiyan Zhang, Zhimu Zhou, Junqi Gao, Dong Li, Jiafei Lyu, Zhouyi Qian, Biqing Qi, Xiu Li, et al. Genprm: Scaling test-time compute of process reward models via generative reasoning. arXiv preprint arXiv:2504.00891, 2025.

---

### Meta-Review · Area_Chair_tK6m · 2026-01-02

**Summary:**

This paper proposes conditional reward modeling (CRM) that frames LLM reasoning as temporal process such that the reward of each reasoning step is not only conditioned on the preceding steps but also linked to the final outcome of the trajectory to overcome the traditional PRMs’ weaknesses in inter-step dependencies and the alignment between process rewards and final outcome.


In the initial reviews, the reviewers are mainly concerned about 1) comparison with latest PRMs and verifiable-reward models; 2) limited scalability and applicability; 3) computational overhead; 4) lack of some important clarifications and discussions

**Reviewer Concerns:**

During rebuttal, multiple reviewers have acknowledged that their concerns are addressed. The final evaluations are all positive. After reading through the paper and all the comments, the AC agrees with the reviewers and recommends accept.

**Reviewer Scores:**

The final scores would be 6, 6, 6, 8,

---

### Decision · Program_Chairs · 2026-01-26

Accept (Poster)